# Oxidative Stress in Breast Cancer: A Biochemical Map of Reactive Oxygen Species Production

**Lyudmila V. Bel'skaya** *[ID] and **Elena I. Dyachenko**

Biochemistry Research Laboratory, Omsk State Pedagogical University, 644099 Omsk, Russia; olseya-120@mail.ru
* Correspondence: belskaya@omgpu.ru

**Abstract:** This review systematizes information about the metabolic features of breast cancer directly related to oxidative stress. It has been shown those redox changes occur at all levels and affect many regulatory systems in the human body. The features of the biochemical processes occurring in breast cancer are described, ranging from nonspecific, at first glance, and strictly biochemical to hormone-induced reactions, genetic and epigenetic regulation, which allows for a broader and deeper understanding of the principles of oncogenesis, as well as maintaining the viability of cancer cells in the mammary gland. Specific pathways of the activation of oxidative stress have been studied as a response to the overproduction of stress hormones and estrogens, and specific ways to reduce its negative impact have been described. The diversity of participants that trigger redox reactions from different sides is considered more fully: glycolytic activity in breast cancer, and the nature of consumption of amino acids and metals. The role of metals in oxidative stress is discussed in detail. They can act as both co-factors and direct participants in oxidative stress, since they are either a trigger mechanism for lipid peroxidation or capable of activating signaling pathways that affect tumorigenesis. Special attention has been paid to the genetic and epigenetic regulation of breast tumors. A complex cascade of mechanisms of epigenetic regulation is explained, which made it possible to reconsider the existing opinion about the triggers and pathways for launching the oncological process, the survival of cancer cells and their ability to localize.

**Keywords:** breast cancer; oxidative stress; ROS; glycolysis; amino acids; hormones; inflammation; genetic and epigenetic regulation

## 1. Introduction

One of the most significant damaging environmental factors acting on biological macromolecules is reactive oxygen species (ROS) [1]. Such forms of oxygen are induced by a variety of physical, chemical and biological factors: ultraviolet and ionizing radiation, the presence of chemical mutagens and carcinogens, as well as impaired aerobic cellular metabolism [2]. The high reactivity of ROS under certain conditions makes them extremely toxic to biological systems at the molecular, cellular and organismal levels of organization [1,2].

An increase in the level of ROS above the limit dictated by antioxidant protection leads biological systems to a state of "oxidative stress", which is accompanied by lipid peroxidation, mutagenic changes in DNA, protein modifications and various pathological processes [3]. On the one hand, a certain physiological level of ROS plays an important regulatory role, participating as specific signaling molecules in the regulation of metabolic processes, gene expression, and the functioning of the immune, endocrine and other physiological systems [4]. On the other hand, the damaging effect of ROS accompanies the development of many diseases, including neurodegenerative, inflammatory, infectious, autoimmune, etc. ROS plays an important role in the processes of aging, mutagenesis, carcinogenesis and teratogenesis [5].

Oxidative stress can cause oxidative damage to lipids, proteins and DNA, including deletions, chromosomal rearrangements, mutations of vital tumor suppressor genes or proto-oncogenes, and also contribute to the stimulation of the Akt/PI3K/mTOR signaling pathway, which are hallmarks of cancer development and progression [6,7]. Disturbances in the redox balance are inherent in the development of many malignant neoplasms, including breast cancer [8]. Breast carcinogenesis is a multistep process, starting with hyperplasia and progressing through atypical hyperplasia to in situ and invasive carcinoma [9]. It is known that in early-stage breast cancer, there is a simultaneous loss of mismatch repair proteins 2 and 6 (MSH2 and MSH6) compared to benign hyperplasia and carcinomas in situ [9]. The expression of 8-hydroxy-2′-deoxyguanosine (8-OHdG), 4-hydroxynonenal (HNE) and nitrotyrosine was more pronounced in early-stage cancer compared to hyperplasias and in situ lesions. Thus, ROS levels are pathologically elevated in breast cancer [10].

Aggressive behavior caused by the production of ROS occurs due to inactivation of tumor suppressors, as well as genomic instability, influencing stem cell phenotypes and chronic inflammation [11]. Activation of ROS itself can occur due to a response to the induction of certain signaling pathways, for example, the epidermal growth factor receptor (EGFR) pathway [12]. An increase in ROS can also occur due to the expression of some oncogenes, such as Ras, myc and telomerase, or, on the contrary, due to the loss of tumor suppressor genes, for example, p53, p21 and PTEN, which leads to cell aging or exit from apoptosis [13]. In addition, ROS can cause overactivation of phosphatidylinositol 3-kinase (PI3K) eAkt by suppressing the activity of the upstream negative regulator phosphatase and tensin homolog (PTEN) in tumor cells [14]. Ouyang et al. found that oxidative stress is associated with the stimulation of cancer cell apoptosis, resulting in disruption of energy metabolism and release of mitochondrial cytochrome C [15]. CYP2E1 influences ROS production, regulates autophagy, stimulates endoplasmic reticulum stress and suppresses the metastatic potential of two metabolically different breast cancer cell lines: triple-negative MDA-MB-231 and estrogen receptor-positive MCF7 [16].

Many antioxidant systems present in living organisms have been discovered. These include superoxide dismutase, catalase, glutathione peroxidase, etc. [17], various proteins, such as albumin, ceruloplasmin and ferritin [18], many compounds of relatively small molecules, such as ascorbic acid, α-tocopherol, β-carotene, ubiquinol-10, glutathione (GSH), methionine, uric acid, bilirubin and hydroxytyrosol, as well as some hormones such as estrogen, angiotensin and melatonin [19–21]. To maintain homeostatic equilibrium, cells modulate mitochondrial metabolism by relying on their internal antioxidant mechanisms [22]. Tumor cells produce more ROS compared to normal cells, thereby facilitating their proliferation and metastasis [23].

In this review, we systematized information about the metabolic features of breast cancer directly related to oxidative stress. We demonstrated that the process of ROS formation could have a wide variety of triggers and a large cascade of reactions. The information we have systematized allows us to take a comprehensive look at the role of oxidative stress in the occurrence and development of breast cancer and to evaluate the contribution of each link. The mechanisms that trigger carcinogenesis are described in sufficient detail, and up-to-date information on possible new therapeutic targets is provided.

## 2. Biochemical Aspects of Breast Cancer and Oxidative Stress

### 2.1. Glycolysis and Oxidative Stress in Breast Cancer

Glucose metabolism produces pyruvate through glycolysis and nicotinamide adenine dinucleotide phosphate (NADPH) in the pentose cycle, which acts as hydroperoxide detoxification against the lethal effects of ROS production during respiration (Figure 1) [24]. When pyruvate reacts with hydrogen peroxide ($H_2O_2$), pyruvate is decarboxylated to acetic acid and $H_2O_2$ is converted to $H_2O$. Thus, pyruvate exhibits antioxidant properties [25]. Glutathione peroxidase (GPx) effectively reduces $H_2O_2$ and lipid peroxides to water and lipid alcohols, respectively, and in turn oxidizes glutathione to glutathione disulfide. Thus, increased consumption of glucose by a cancer cell is necessary to enhance

glucose metabolism, resulting in the protection of the cell from the overproduction of internal ROS. [26]. This strong dependence of tumor cells on glucose to detoxify ROS is an attractive therapeutic target that kills cancer cells while bypassing healthy ones [27]. Under conditions of glucose deprivation, ROS production by the mitochondrial respiratory chain promotes oxidative stress and apoptosis [28]. Glucose deprivation has been shown to reduce the products of glucose metabolism, leading to total glutathione (GSH) depletion and increased ROS [29].

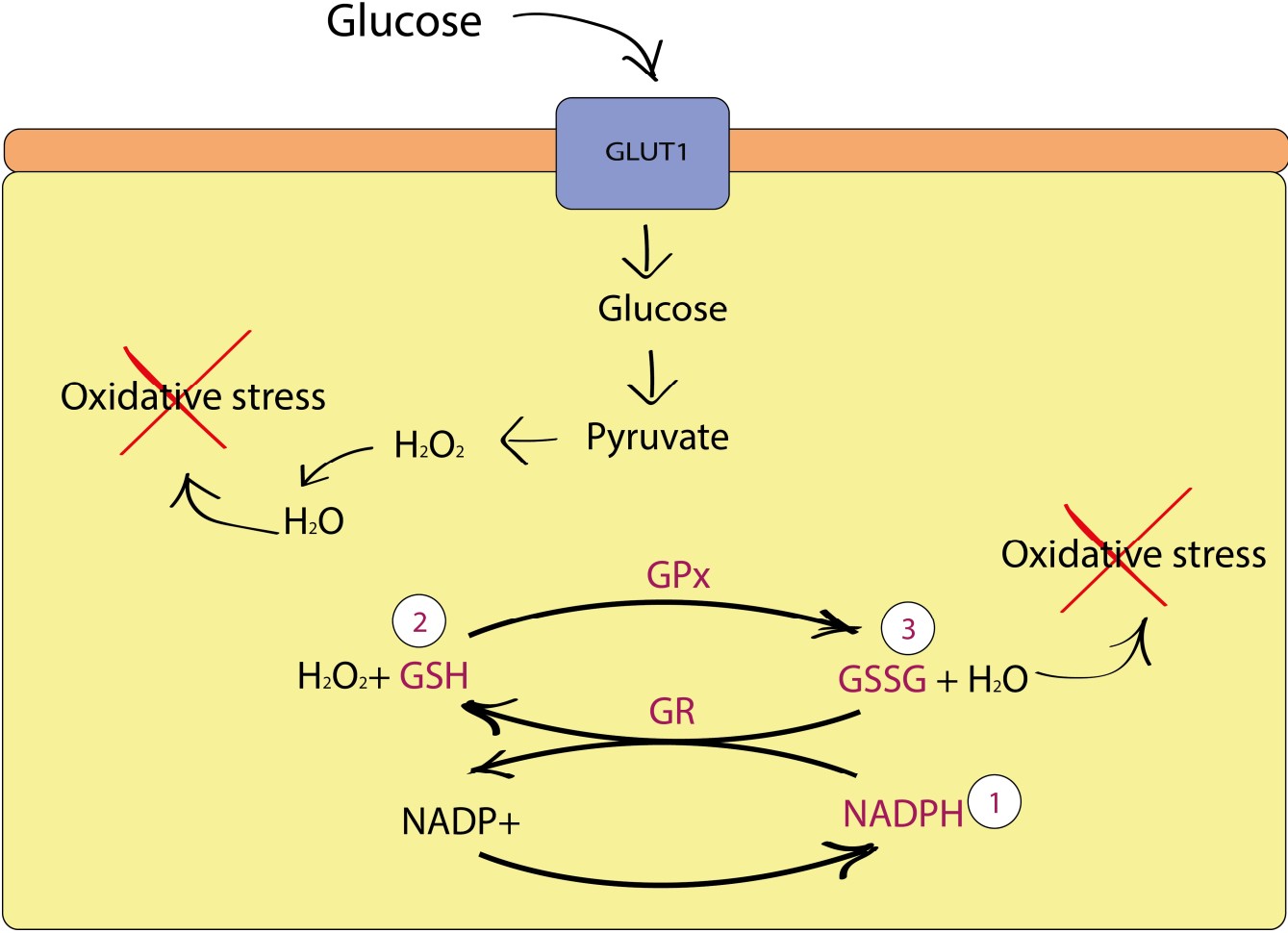

**Figure 1.** Antioxidant properties of pyruvate. Glucose, entering the cell, after a series of reactions, is metabolized into pyruvate. Pyruvate reacts with $H_2O_2$ to form water and carbonic acid. NADP also interacts with $H_2O_2$ under the action of glutathione peroxidase, providing electrons for the reduction of glutathione disulfide. GLUT1—Glucose transporter type 1; NADP—nicotinamide adenine dinucleotide phosphate; NADPH—nicotinamide adenine dinucleotide phosphate hydrogen; GSH—glutathione; GSSG—glutathione disulfide; GPx—glutathione peroxidase.

Glucose provides electrons for hydroperoxide metabolism through pentose cycle activity to regenerate NADPH, which serves as an electron donor for GSH- and thioredoxin (Trx)-dependent peroxidase activity, and through glycolysis to produce pyruvate, which can directly respond to hydroperoxide detoxification via decarboxylation reactions [30,31]. Overall, inhibition of glycolytic and pentose cycle activities coupled with inhibition of Trx metabolism may provide a promising strategy for selectively sensitizing human cancer cells to oxidative stress-induced cell killing [32].

As a protective response against ROS overproduction, cancer cells have acquired changes in metabolic pathways involved in maintaining redox homeostasis. These changes include upregulation of the pentose phosphate pathway or glutamine uptake, which

supports the synthesis of NADPH and GSH, respectively [32,33]. Another antioxidant defense system for cancer cells is the thioredoxin (Trx) system [34]. Thioredoxin reductase (TrxR) is commonly overexpressed in cancer cells [35] and is involved in antioxidant activity, drug resistance, anti-apoptosis, metabolism, and cancer relapse [36]. For example, TrxR/Trx directly neutralizes ROS or affects other antioxidants or antioxidant enzymes [37] and may increase resistance to chemotherapeutic drugs and inhibit apoptosis [38]. TrxR/Trx regulate glyceral dehyde-3-phosphate dehydrogenase (GAPDH) by altering the glycolytic pathway [39]. Moreover, TrxR/Trx activate p53 and hypoxia-inducible factor 1$\alpha$ (HIF-1$\alpha$), affecting cell proliferation [40], as well as angiogenesis and metastasis [41,42], which may have applications in the development of therapeutic strategies [43–45].

It has been shown that high levels of glycolytic activity have been observed in breast cancer tissues and cell lines [46,47]. High levels of glycolysis are associated with increased cell proliferation [48]. Consistent with this, increased expression of the major glucose transporter in cancer cells, glucose transporter 1 (GLUT1) (also known as SLC2A1), has been associated with increased proliferation and consequent poor prognosis [49]. In particular, the glucose transporters GLUT1 and GLUT3 are more highly expressed in breast cancers, with lower differentiation [50]. Increased glucose uptake through the GLUT1 transporter is observed in triple-negative breast cancer [51], which correlates with a poor prognosis [52]. A number of authors have considered GLUT1 inhibition as a possible cancer treatment, but this remains controversial to date [53]. Increased glycolytic activity is also explained by the greater activity of some enzymes in this pathway, such as hexokinase 2 [54]. Elevated levels of phosphofructokinase have been found in breast cancer [55]. Overexpression of pyruvate kinase M2 is associated with decreased survival and an increased risk of metastasis in breast cancer [56].

The reasons for the low levels of tricarboxylic acid (TCA) cycle activity and the mechanism by which pyruvate does not enter the TCA cycle in emerging tumors remain incompletely understood. In the case of breast cancer, this may be explained by low levels of expression of the pyruvate dehydrogenase protein X (PDHX), a structural component of the PDH complex, an enzyme that controls the flow of metabolites from glycolysis to TCA. Reduced PDHX expression is associated with poor survival in breast cancer [57]. Some subtypes of breast cancer differ in the level of expression of certain enzymes involved in TCA. Thus, in HER2-positive tumors, the expression of succinate dehydrogenase subunit A is significantly higher than in luminal A-like tumors [58]. Another pathway for glucose oxidation, in addition to glycolysis and TCA, is the pentose phosphate pathway (PPP). In cancer, there is an increase in metabolism along this pathway. One molecule of glucose produces one molecule of ribulose-5-phosphate and two molecules of NADPH. NADPH is primarily used for fatty acid synthesis and the reduction of glutathione. Ribulose-5-phosphate is converted to ribose 5-phosphate, which is used for the synthesis of nucleotides and nucleic acids. Thus, PPP connects the metabolism of carbohydrates and fatty acids, anaplerosis, nucleotide synthesis and antioxidant protection, depending on the individual metabolic needs of the cell. Cancer cells have a high demand for NADPH to neutralize the radical oxidized species (ROS), which they produce in large quantities [59]. Since PPP is very important for tumor cell proliferation, it is clear why high levels of some PPP enzymes, such as glucose-6-phosphate dehydrogenase and transketolase, are directly correlated with poor evolution in breast cancer [60]. In addition, some PPP enzymes are predominantly expressed in HER2-positive tumor subtypes. It is assumed that it is for this molecular subtype of neoplasm that the main and most significant pathway of glucose metabolism is the pentose phosphate pathway [61].

In breast cancer, it has been shown that symbiosis or crosstalk occurs between oxygenated and non-oxygenated areas of the tumor, thereby compensating for nutrient and oxygen deficiencies (Figure 2). In areas of increased hypoxia, HIF1$\alpha$ and expression of the lactate transporter MCT4 (monocarboxylate transporter-4) are induced. Under conditions of reduced oxygenation, cells with high glycolytic activity that produce significant amounts of lactate release it into the interstitium through the MCT4 transporter. Lactate is then

taken up by cancer cells in well-oxygenated areas that are HIF1α negative and express the MCT1 transporter. Cancer cells, saturated with oxygen, preferentially use lactate rather than glucose. It has been suggested that cells from oxygenated areas have a lower glucose requirement than cells from poorly oxygenated areas, thereby maintaining the well- and poorly oxygenated areas of the tumor [62]. MCT4 is the major exporter of L-lactate from cells and is a marker of oxidative stress and glycolytic metabolism.

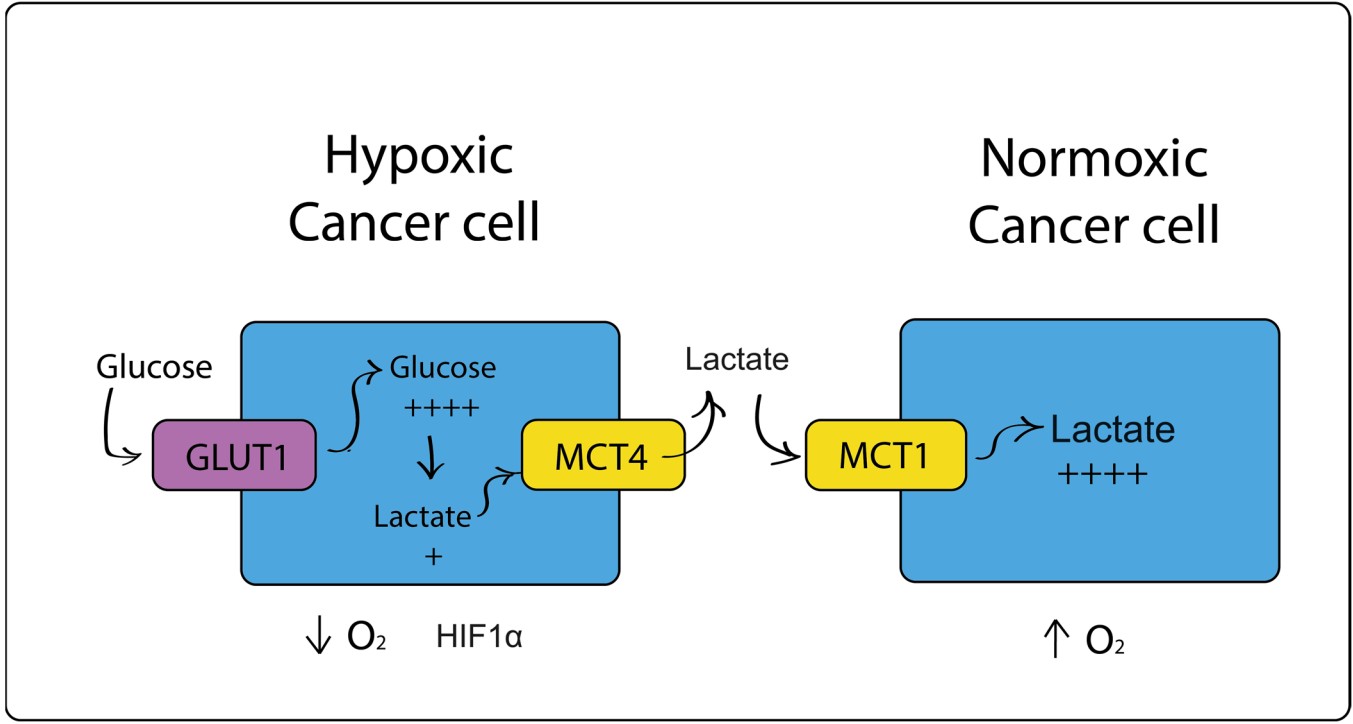

**Figure 2.** Differences in lactate consumption between well-oxygenated and poorly oxygenated cancer cells. Cells from well-oxygenated areas consume more lactate. Conversely, cells with low oxygenation consume more glucose. HIFα—hypoxia-inducible factor-1; MCT 1, MCT4—monocarboxylate transporter 1, monocarboxylate transporter 4; the sign "++++" reflects a high concentration in the cell; the sign "+" reflects a low concentration in the cell.

Glucose and glutamine are two essential nutrients for cancer cells [63]. Recently, solute transporter 7 family member 11 (SLC7A11), also called xCT, was found to be involved in glucose-dependent pathways [64]. It has been shown that breast cancer cells with high xCT expression are more susceptible to glucose deprivation [65], which is promoted by increased ROS levels. In addition, during glucose deprivation, SIRT3 may play a protective role in ROS levels, particularly mtROS. SIRT3 transcription can be activated by oxidative stress or nutrient deprivation [66]. SIRT3 activates manganese superoxide dismutase (MnSOD) by deacetylating MnSOD-K68 to reduce ROS levels [66] and deacetylates isocitrate dehydrogenase 2 and enzymes in the TCA cycle and electron transport chain, regulating ATP production, fatty acid oxidation, glucose oxidation and mitochondrial biogenesis [67]. Loss of SIRT3 can lead to increased ROS levels, genomic instability and intracellular metabolism, with increased glycolysis and decreased oxidative phosphorylation [68].

Oncometabolites such as fumarate, D-2-hydroxyglutarate and succinate accumulate in tumors and regulate epigenetic changes [69]. In breast cancer cells, both ketones and lactate are associated with increased stemness and oxidative stress. Mutations in the MYC, TP53 and MTOR genes have been observed to promote chemoresistance in breast cancer [70]. Some oncogenes, such as NF-kB, enhance cell proliferation by altering glycolysis [71]. Aerobic glycolysis describes the use of glucose through glycolysis rather than oxidative phosphorylation by tumor cells to meet their high energy demands (the

"Warbrug effect"). Cancer cells maintain their high glycolysis rate by overexpressing the glucose transporter and monocarboxylate transporter [72]. At the same time, migration and invasion are enhanced due to the acidic extracellular tumor environment, which promotes tumor growth [73]. Overexpression of hexokinase II and GLUT-1 in various cancers has been found to inhibit apoptosis. It has been shown that for a better antitumor response in TNBC, there must be a glucose deficiency [74,75]. Overall, glucose metabolism plays a vital role in understanding breast cancer progression.

### 2.2. Amino Acids and Oxidative Stress in Breast Cancer

In addition to glucose, many studies have identified other sources of carbon and nitrogen required for tumor growth, including glutamine, proline, branched-chain amino acids and serine [76].

Breast cancer cells have a high need for glutamine metabolic products as the main sources of antioxidant protection. Glutamine is converted into glutamic acid by the enzyme glutaminase. This enzyme is overexpressed in TNBC tumors compared to HER2-enriched and luminal subtypes [77]. Exogenous glutamine has been shown to be essential for TNBC cell survival [78]. Cancer cells of the luminal subtype are less dependent on the supply of exogenous glutamine since they have the ability to independently synthesize this amino acid due to the enzyme glutamine synthase [79].

As stated above, an important glutamine product is glutamic acid or glutamate, which is a precursor to GSH, whose role is to enhance the antioxidant response in cancer cells [80]. In addition, glutamine provides a nitrogen source for nucleotide synthesis and supports activation of the mechanistic target of the rapamycin (mTOR) signaling pathway. This pathway regulates protein synthesis, metabolism, tumor growth and the ability to metastasize in cancer [81]. The TNBC and HER2-positive subtypes were found to actively consume glutamine and have an enhanced glutaminolysis reaction [82]. The glutaminase inhibitor CB-839 has shown efficacy against TNBC in preclinical studies [83].

Since extracellular glutamine cannot simply diffuse into cells, transmembrane glutamine transport is an active process mediated by transporters that belong to four families: SLC1, SLC6, SLC7 and SLC38 [84]. SLC1A5 and SLC7A5 are known to be upregulated in TNBC human breast cancer cells and their inhibition reduces tumor cell growth in TNBC [85,86]. SLC6A14 is a drug target for the treatment of ER+ breast cancer [87]. Morotti et al. showed that system A transporters (SNAT1 and SLC38A2) are highly expressed in breast cancer cell lines [88]. The authors showed that the antitumor effect of SLC38A2 is partially mediated by oxidative stress and that high expression of this protein correlates with poor survival. Glutamine sensitivity is generally more common in TNBC subtype cell lines [79], but estrogen-positive cell lines, such as MCF7, can also exhibit glutamine sensitivity [89]. These data suggest that early metabolic responses of cancer cells to microenvironmental stresses (e.g., nutrient deprivation) may be common across subtypes and extend beyond the given genetic background of that cell line.

The SLC38A3 protein is overexpressed in TNBC cell lines [90]. As a neutral amino acid transporter, SLC38A3 increases the influx of glutamine, asparagine and alanine and enhances the subsequent formation of glutamate and aspartate in breast cancer. Glutamine is a precursor to reduced GSH [91], present in cells at concentrations 10–100 times higher than its oxidized form GSSG [92]. The vulnerability of cancer cells to oxidative stress increases with GSH deficiency, or the GSH/GSSG ratio decreases [93]. SLC38A3 is required for the regulation and maintenance of cellular GSH levels and GSH/GSSG ratio in breast cancer. High levels of SLC38A3 have been shown to increase total β-catenin levels in breast cancer [94] by stabilizing it [95]. Glutamate may serve as a nitrogen source for the production of other amino acids by aminotransferases in cancer [96,97]. Increased cellular amino acid content can stimulate mTOR, which in turn can inactivate Gsk3β [98]. Lactate levels are reduced in breast cancer cells when SLC38A3 is suppressed [99]. In addition to protein synthesis, nucleotide synthesis is an amino acid-dependent process [100]. For the synthesis of purines and pyrimidines, glycine, glutamine and aspartate are used as

carbon and nitrogen donors, which is important for supporting cancer proliferation and therapeutic resistance [101].

Under conditions of increased ROS levels, the production of endogenous L-cysteine (L-Cys) is insufficient for GSH synthesis, which requires the uptake of L-Cys, predominantly in its disulfide form of L-cystine (CSSC), through the xCT transporter, which consists of two subunits: SLC7A11 and SLC3A2 (Figure 3) [102]. Cyst(e)ine is a key precursor of GSH synthesis, which protects cancer cells from oxidative stress [103]. One of the factors suppressing GSH production is the absence of cysteine in the cellular environment [104]. Increased lysosomal stores of cyst(e)ine, which is released by cystinosin (CTNS) to maintain GSH levels and buffer oxidative stress in breast cancer cells, occurs through activation of the major facilitator superfamily domain containing 12 (MFSD12). He et al. found that mTORC1 regulates MFSD12 by directly phosphorylating residue T254, whereas inhibition of mTORC1 enhances lysosomal acidification, which activates CTNS, which as a switch modulates lysosomal cyst(e)ine levels in response to oxidative stress (Figure 3). The MFSD12-T254A mutant inhibits MFSD12 function and suppresses tumor progression, whereas overexpression of MFSD12 worsens the prognosis of breast cancer patients. Cramer et al. proposed the administration of an engineered, pharmacologically optimized human cyst(e)inase enzyme that mediates sustained depletion of the extracellular pool of L-Cys and CSSCs and does not result in overt toxicity in mice even after several months of continuous treatment [105].

SLC38A5 increased in TNBC cell lines compared to other amino acid transporters, such as SLC38A1 and SLA38A2 [106]. Increased levels of SLC38A5 promote cell proliferation, cell cycle progression and the inhibition of cell apoptosis in TNBC cell lines [107]. Overexpression of SLC38A5 promotes breast cancer cell viability through glutamine metabolism. The amino acid-dependent Na+/H+ exchange activity of SLC38A5 promotes macropinocytosis, which may contribute significantly to amino acid nutrition in cancer cells [108]. Recently, it was shown that SLC38A5 could transport selenomethionine (Se-Met), the most common dietary source of selenium. Since the selenoenzyme GPX4 is a critical component of the antioxidant machinery in TNBC cells, the possible involvement of SLC38A5 in the delivery of selenium to cancer cells in the form of Se-Met may represent a novel function of the transporter in promoting cancer growth [109]. Fan et al. showed that Se-Met is an activator of Nrf2, an important transcription factor associated with the antioxidant mechanism [110].

Stimulation of cell growth occurs under the influence of insulin-like growth factor (IGF), partly due to increased amino acid uptake [111]. Cystine uptake into cells occurs through the cell surface subunit of the xC transport system, encoded by xCT (SLC7A11). Cystine is involved in a key step in GSH synthesis and cellular redox control [112]. Yang et al. showed a regulatory role of IGF-I [113] in cystine uptake and cellular redox status upon xCT activation in breast cancer cells that are estrogen receptor positive (ER+). This occurs through a mechanism based on the IGF-I receptor substrate (IRS-1). A study was conducted using sulfasalazine (SASP). IGF-I-mediated breast cancer cell proliferation was suppressed when xCT expression was attenuated or completely blocked by SASP. Importantly, SASP sensitized breast cancer cells to IGF receptor inhibitors (IGF-IR) in a manner that was modified by the ROS scavenger N-acetyl-L-cysteine. Thus, stimulation of ER+ breast cancer cell proliferation is regulated by IGF-I, which controls the function of the xCT transporter to protect cells from ROS through IRS-1.

Serine, a nonessential amino acid that can be synthesized in the body, is another important amino acid in breast cancer [114]. 3-Phosphoglycerate dehydrogenase (PHGDH) is the first enzyme involved in serine synthesis. It is overexpressed in breast cancer [115] and breast cancer cell lines [116] and mainly in subtypes of breast cancer that proliferate more significantly, such as ER-negative cells [117]. High expression of serine metabolism-related enzymes, such as phosphoglycerate dehydrogenase (PHGDH) and phosphoserine phosphatase (PSPH), is observed in aggressive cancer subtypes [118]. Its activity is increased in TNBC, estrogen receptor-negative (ER) breast cancer and metastatic variants of ER-negative breast cancer cells. Importantly, its high expression in these tumors is associated with a poor

prognosis [115]. PHGDH overexpression and serine synthesis are associated with tumor growth for several reasons. Serine nourishes protein synthesis pathways and stimulates one-carbon metabolism, which includes nucleic acid synthesis through folate, antioxidant defense and methylation reactions [119]. Enhancement of the serine pathway may promote the TCA cycle by complementing intermediates in the TCA cycle. This suggests that the serine pathway is required for tumor cell growth [120]. Increased expression of PSPH promotes the synthesis of the oncometabolite 2-hydroxyglutarate (2HG) [120]. To date, there are no FDA-approved drugs against PHGDH. However, experimental inhibitors are being improved for future clinical trials [121].

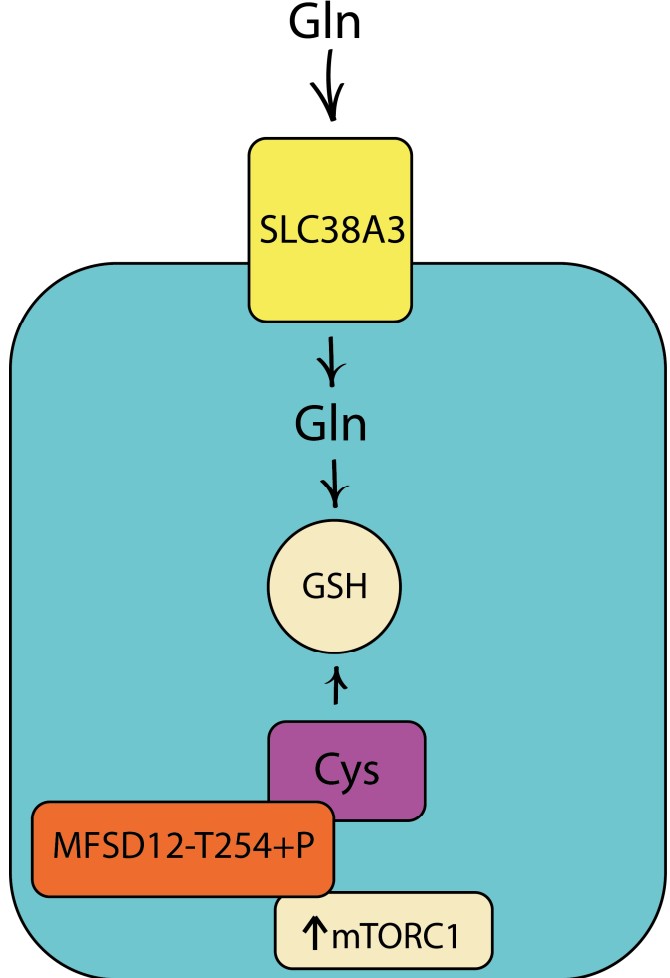

**Figure 3.** Glutathione synthesis. Glutamine is transported from the extracellular space into the cell using a transmembrane protein SLC38A3 inside the cell, followed by conversion to glutathione. Glutathione is also formed after mTORC1 phosphorylates MFSD12, which leads to the accumulation of cysteine in lysosomes for subsequent transformation into glutathione. Gln—glutamine; SLC38A3—sodium-coupled neutral amino acid transporter 3; GSH—glutathione; Cys—cysteine; MFSD12—major facilitator superfamily domain containing 12; mTORC1—mammalian target of rapamycin complex 1; P—phosphorus.

Proline metabolism is important for regulating the survival and death of various types of cancer cells [122]. Proline dehydrogenase (PRODH), an enzyme that catalyzes proline catabolism, and proline degradation products by PRODH, such as ATP and ROS, are known to play a critical role in cancer progression [122]. Human mitochondrial pyrroline 5-carboxylate reductase (PYCR) is a housekeeping enzyme that catalyzes the reduction of Δ1-pyrroline 5-carboxylate to proline [123]. This enzymatic cycle plays a key role in amino

acid metabolism, intracellular redox potential and mitochondrial integrity [124]. Higher PYCR1 mRNA levels were significantly associated with poor survival in breast cancer patients, regardless of estrogen receptor status [123]. Proline oxidase, also known as proline dehydrogenase (POX/PRODH), is a key enzyme in the proline metabolic pathway and plays a vital role in tumorigenesis [125,126]. POX has been shown to promote cell survival under chemotherapeutic stress and may serve as a potential target for the treatment of TNBC [127]. Moreover, the absence of estradiol inhibits collagen biosynthesis, thereby providing proline for PRODH to induce apoptosis by generating ROS in TNBC [128].

Glycine promotes rapid proliferation of breast cancer cells [129,130]. Glycine has been shown to be consumed by rapidly proliferating cells and released by slowly proliferating cells. This suggests that the demand for glycine may exceed the capacity for endogenous synthesis in rapidly proliferating cancer cells [130]. Glycine is the third amino acid added to $\gamma$-glutamylcysteine, the condensation product of glutamate and cysteine, to form GSH [131]. Glycine deficiency decreases the synthesis of GSH and promotes the production of ROS. Glycine can be transported into cells via the specific glycine transporters GLYT1 and GLYT2, as well as by a variety of non-specific amino acid transporters: SLC6A14, SLC36A1, SLC36A1, SLC38A2 and SLC38A4 [132]. Recently, SLC6A14 and SLC38A5 have been shown to be upregulated in various cancers and mediate the influx of glutamine, serine, glycine and methionine into cancer cells and are suitable for the rapid proliferation of cancer cells [133]. Thus, these two amino acid transporters play a critical role in promoting the survival and growth of cancer cells and therefore represent new, still largely unexplored targets for cancer therapy.

In general, the transport and internal synthesis pathways of cysteine, serine, glutamine and, to some extent, glycine seem to be the most interesting targets for the development of new redox drugs [134]. Targeting the amino acid transport systems (xCT, ASCT2 and SNAT) is promising given that import of these semi-essential amino acids is not required in normal cells, while they are absolutely essential for cancer cell survival.

### 2.3. Heavy Metals and Oxidative Stress in Breast Cancer

Ferroptosis is a unique cell death mechanism that is dependent on the presence of iron and coexists with increased oxidative stress and lipid peroxidation [135]. In cases of resistance to apoptotic pathways, ferroptosis is an excellent alternative for overcoming the suppressed therapeutic response to anticancer drugs [136]. Bakar-Ates et al. showed that the combination of cucurbitacin B and erastin was able to activate ferroptotic cell death in MCF-7 and MDA-MB-231 breast cancer cells through suppressing antioxidant function, increasing lipid peroxidation and altering the expression of iron regulatory proteins [137]. Ferroptosis is based on complex and abnormal biochemical processes, including the metabolism of amino acids, ions and polyunsaturated fatty acids, as well as the biosynthesis of glutathione, phospholipids, NADPH, coenzyme Q10, etc. [104]. Ferroptosis can be caused by ROS [138]. The iron-mediated Fenton reaction causes oxidative stress and lipid peroxidation, causing ferroptosis (Figure 4) [139]. When divalent iron reacts with $H_2O_2$, it gives up an electron and produces a hydroxyl radical, causing lipid peroxidation and triggering a cascade of reactions leading to cell death [140]. Reactive nitrogen species (RNS) also promote ferroptosis [141], particularly nitric oxide (NO) and peroxynitrite (ONOO-) [142]. RNS forms lipid peroxides in plasma membranes and intracellular organelles [143]. Glutathione peroxidases (GPX), in particular GPX4, play a decisive role in the regulation of oxidative stress [144]. GSH, as a substrate for GPX4, protects lipids from peroxidation and prevents ferroptosis [144]. Using the example of MDA-MB-231 breast cancer cells, it was shown that 18-$\beta$-glycyrrhetinic acid (GA) suppressed the expression of SLC7A11 of the xc- system, reduced the level of GSH, inhibited GPX activity and induced ferroptosis [145].

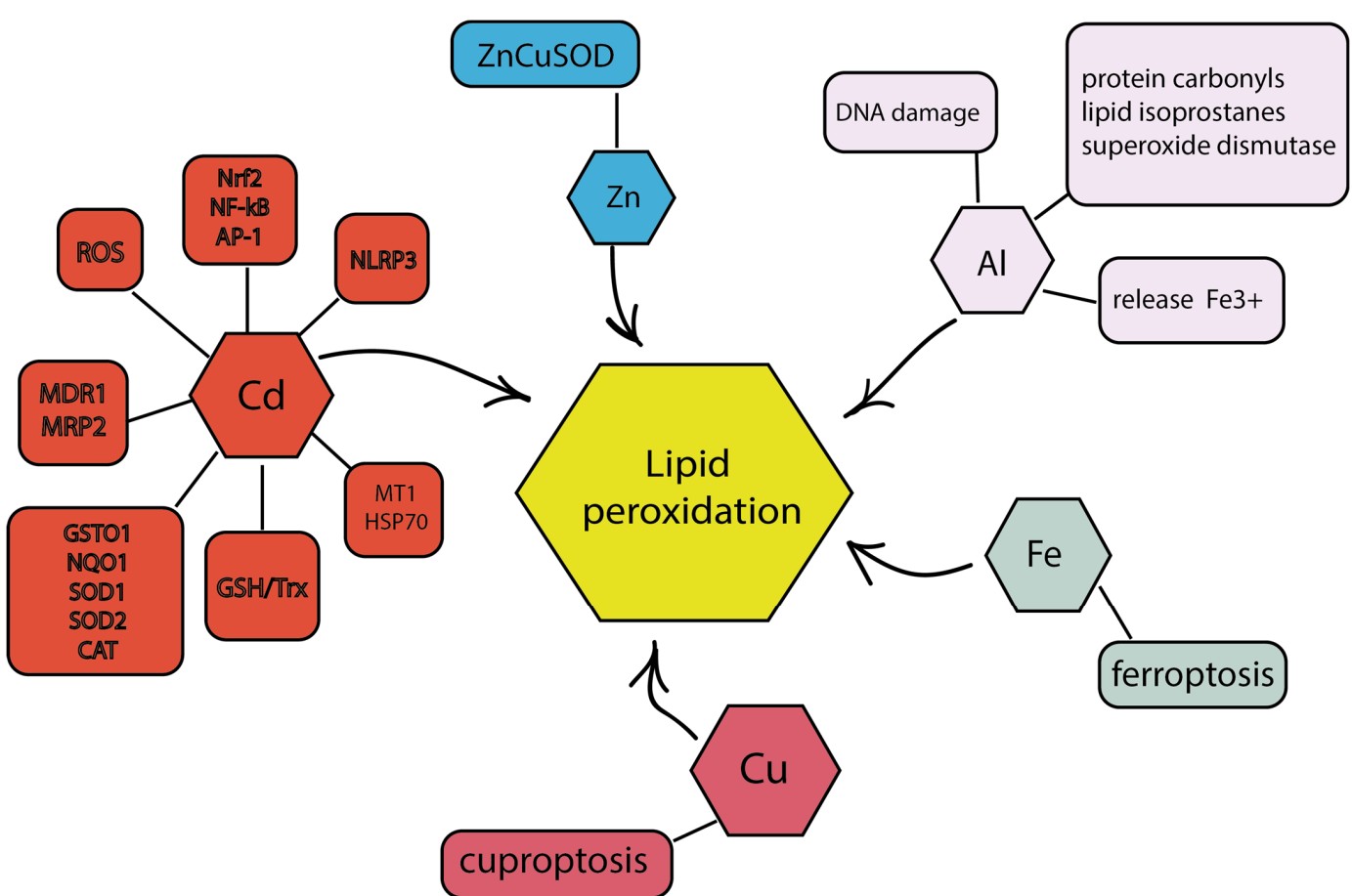

**Figure 4.** The main biochemical reactions of inflammatory processes in which metals take part. The main general effect of metals on oxidative stress is lipid peroxidation. Nrf2—nuclear factor erythroid 2-related factor 2; NF-kB—nuclear factor kappa-light-chain-enhancer of activated B cells; AP-1—activating protein-1; NLRP3—NOD-, LRR-, and pyrin domain-containing protein 3; MT1—melatonin receptors 1; HSP70—heat shock protein 70; GSH/Trx—glutathion/thioredoxin; MDR1—multidrug resistance mutation 1; MRP2—multidrug resistance protein 2; ROS—reactive oxygen species; GSTO1—glutathione S-transferase omega-1; NQO1—NAD(P)H quinone dehydrogenase 1; SOD1—superoxide dismutase 1, SOD2—superoxide dismutase 2; CAT—catalase.

Dysregulation of copper content can lead to the overproduction of ROS, which can act as a precursor to the development of neoplastic transformation and the formation of metastases [146,147]. Copper is known to stimulate TNBC metastasis by modulating mitochondrial oxidative phosphorylation [148]. Cuproptosis, a recently identified form of regulatory cell death, results from excessive copper accumulation and is closely associated with breast cancer proliferation and metastasis [149,150]. Copper is involved in cell proliferation, angiogenesis and metastasis and influences the progression of breast cancer [151]. Due to their wide range of catalytic functions and electrochemical activity [152], copper complexes can have tumor inhibitory effects by inducing ROS production, glutathione depletion, proteasome inhibition and DNA damage [153]. As an example, it has been shown that copper(II)-hydrazone complex (CuHL1) and copper complex of plumbagin (Cu-PLN) can effectively inhibit the growth of TNBC cells by promoting cuproptosis (Figure 4). CuHL1 suppresses the expression of dihydrolipoamide S-acetyltransferase (DLAT) through copper binding-induced protein lipoylation [154], while Cu-PLN can cause DNA damage and ROS production and affect mitochondrial metabolism [155]. In addition, Li et al. developed a novel copper-coordinated covalent organic scaffold that promotes ROS production, induces immunogenic tumor cell death, increases CD8+ T cell infiltration and

improves the response rate to immune checkpoint inhibitor therapy [156]. In preclinical studies, copper has been shown to play various roles in cancer progression, including angiogenesis, tumor growth and metastasis [157].

Zinc acts as a cofactor for more than 300 enzymes, such as transferases, hydrolases and isomerases. In addition, the metal is involved in the regulation of many physiological processes. Thus, it takes part in the anti-inflammatory response as well as antioxidant and immune reactions, and can also indirectly regulate the ability of cells to apoptosis [158]. Zinc is part of many vital proteins. For example, it contains a zinc finger transcription factor that regulates gene activity. It is a cofactor of the antioxidant enzyme copper/zinc superoxide dismutase (CuZnSOD), which catalyzes the conversion of superoxide to oxygen and $H_2O_2$. It can be assumed that higher zinc levels in the body may lead to a better prognosis for breast cancer [159,160]. Overexpression of CuZnSOD has been shown to suppress the growth of breast cancer cells in vitro [161] and reduce the metastatic activity of breast cancer in vivo [162]. It can be assumed that an imbalance in the content of zinc and copper can lead to dysfunction of CuZnSOD, which will lead to ROS accumulation and the triggering of oxidative stress [163].

Aluminum salts accumulate in breast cells/tissues/fluids, mainly due to cosmetics/antiperspirants [164,165] and may be able to alter the expression of iron-binding proteins such as L-chain ferritin and transferrin in the breast cancer microenvironment [166]. Increased iron metabolism can cause the release of ferric iron, which can be reduced to ferrous iron through the aluminum superoxide complex, promoting oxidative damage through the Fenton reaction (Figure 4) [167]. Mannello et al. showed that in breast cancer, nipple aspirate fluid (NAF) contains altered levels of ROS-related compounds and oxidative products (e.g., protein carbonyls, lipid isoprostanes, superoxide dismutase [168]. Celik et al. found a significant positive correlation between high amounts of aluminum and protein carbonyls, i.e., aluminum may be involved in such oxidative stress pathways [169]. It was demonstrated by Sappino et al. that aluminum salts cause damage and double-strand breaks in DNA [170], predisposing human mammary cell lines to proliferative and carcinogenic stress.

Cadmium exposure can cause increased levels of ROS [171], epigenetic changes including DNA methylation and chromatin remodeling [172] and activation of ER$\alpha$ and downstream pathways (Figure 4) [173]. All of this can contribute to the malignant transformation of breast cells [174]. Cadmium has been shown to disrupt critical antioxidant systems based on glutathione or thioredoxin by binding to the sulfhydryl groups of enzymes [175] and interfering with the mitochondrial electron transport chain [176]. In general, the adverse effects of cadmium lead to increased intracellular ROS levels, DNA damage and mutations, cellular dysfunction and necrosis [177]. ROS production in MCF-7 cells is dose-dependent on cadmium concentration [171]. Tang et al. showed that increased ROS generation by cadmium activates the NLRP3 inflammasome pathway, promoting the GSDME-mediated pyroptosis of MDA-MB-231 cells [178]. In addition, cadmium promotes the adaptive response to oxidative stress through activation of the Nrf2, NF-$\kappa$B and AP-1 pathways [179]. Darwish et al. showed that cadmium at concentrations of 10 and 100 $\mu$M induces lipid peroxidation in MCF-7 cells [180], which can cause changes in the assembly, composition, structure and dynamics of lipid membranes [181]. In addition, cadmium disrupts the antioxidant enzyme system by reducing the expression of antioxidant genes (GSTO1, NQO1, SOD1, SOD2 and CAT), detoxification enzymes (MT1 and HSP70) and xenobiotic transporters (ATP-binding cassette transporters MDR1 and MRP2) [180].

## 2.4. Selenium and Oxidative Stress in Breast Cancer

Selenium is well known as an essential trace element and an important component of the antioxidant enzyme glutathione peroxidase (GPx), which is critical for ROS scavenging and maintaining redox balance [182]. Some studies have shown that selenium exposure causes apoptosis of cancer cells through the production of superoxide radicals [183]. Sodium selenite activates oxidative stress by increasing ROS generation and decreasing

mitochondrial membrane potential [45,184]. It is known that selenium can affect the early stages of carcinogenesis, inhibit the proliferation of malignant cells, induce their apoptosis and activate antioxidant enzymes (cytochrome P450, microsomal hydrolase or glutathione transferase) [185].

The physiological function of selenium related to its anticancer effects is associated with selenium-containing proteins, which can be divided into three groups: specific seleno-proteins, non-specific selenium-containing proteins and selenium-binding proteins, which include liver fatty acid binding protein, protein disulfide isomerase and selenium binding protein 1 (SELENBP1) [186].

In particular, SELENBP1 expression is reduced in breast cancer tissues compared to normal controls [187]. Low expression of SELENBP1 in patients with ER+ breast cancer was significantly associated with poor survival. In this case, the expression of SELENBP1 is regulated by estrogen and the effect of inhibition of cell proliferation upon treatment with selenium depends on the high level of SELENBP1 expression. Thus, the expression level of SELENBP1 may be an important marker for predicting survival and the effectiveness of selenium supplementation in breast cancer.

It is assumed that the antioxidant effects of selenium are mediated primarily through selenoproteins, which use selenocysteine residues to catalyze redox reactions in cells [188]. For example, selenophilic cancer cells (breast cancer cells) have higher levels of selenopro-teins, which protect them from ferroptosis [189]. Selenium status generally correlates with levels of pro-inflammatory cytokines in cancer [190]. Selenium is also involved in hormonal carcinogenesis; in particular, selenium in the composition of methyl selenous acid blocks the estradiol-dependent growth of tumor cells in breast cancer and the expression of the ER-$\alpha$ gene [191].

## 3. Oxidative Stress in Breast Cancer

### 3.1. Inflammation and Oxidative Stress in Breast Cancer

Inflammation is accompanied by increased levels of free radicals as reactive oxygen and nitrogen species bind DNA and cause mutations, leading to cellular instability, promoting carcinogenesis [192]. Moreover, the same molecules can trigger multiple pathways involved in further cancer growth [193]. Chronic inflammation may promote angiogenesis [194]. Normal breast tissue consists of hormone-sensitive epithelial cells, adipocytes and an extracellular matrix, which is filled with fibroblasts and macrophages, elements sensitive to inflammatory signals [195]. Adipocytes actively secrete cytokines in response to proinflammatory signals [196].

Oxidative stress is characterized by the presence of persistent free radicals, which leads to the induction of a chronic inflammatory response. This increases the level of tumor necrosis factor $\alpha$ (TNF-$\alpha$) due to macrophage infiltration, dysregulation of interleukin-6 (IL-6) and the production and activation of inflammatory enzymes such as cyclooxygenase 2 (COX2) [197]. COX2 is a potent oxidant that oxidizes adjacent cellular substrates, resulting in DNA damage [198]. Increased levels of COX2 may lead to the depletion of arachidonic acid metabolites, which in turn reduces cell apoptosis [199,200]. Collectively, this worsens the prognosis of breast cancer by reducing drug response and increasing metastasis. Overexpression of COX2 results in increased production of prostaglandin 2 (PGE2), which leads to subsequent signaling through the MAPK, Src and Akt pathways, as well as VEGF and HIF-1$\alpha$ [201]. HIF-1$\alpha$ mediates the link between hypoxia and inflammation in tumors. Activation of the above metabolic pathways leads to the progression of breast cancer [202]. It is also known that the risk of developing breast cancer is reduced by taking non-steroidal anti-inflammatory drugs that inhibit COX2 [203]. What is common in the actions of these inflammatory molecules in breast cancer is that all of these pathways can modulate aromatase production [204]. Aromatase, an isoform of mitochondrial cytochrome P-450 involved in the oxidation of endogenous and exogenous compounds, is vital for the production of estrogens, a critical factor in the etiology and progression of breast cancer [204].

Increased ROS production leads to decreased antioxidant capacity and is associated with cytokine-induced inflammation [205]. Oxidative stress induced by cytokines can induce mammary carcinogenesis by causing genomic instability [193]. IL-6 characterizes the acute-phase response at the site of inflammation and injury, and its level increases with tumor progression [206]. It is known that IL6 levels are increased in the serum and tissues of patients with TNBC [207]. On the other hand, IL6-mediated inflammation is considered a risk factor for TNBC [208]. Inflammation may contribute to the development of more aggressive, treatment-resistant types of breast cancer [209]. IL-8 is an angiogenic chemokine [210] that can activate neutrophils, basophils and T cells during the inflammatory process with a longer half-life than IL-6 [211]. The role of IL-10 is anti-inflammatory [212]; however, several studies have reported a role for IL-10 in breast cancer, as IL-10 mRNA was found to be highly expressed in breast cancer cells [213]. TNF-$\alpha$ is involved in systemic inflammation and can stimulate the acute phase response [214].

ROS are closely associated with chronic inflammation, as evidenced by altered lipid peroxide concentrations in breast cancer [215]. Thus, in invasive ductal carcinoma, free hydroxyl radicals can attack DNA [216]. The hydroxyl radical generated during inflammation is involved in base modifications, including the formation of thymine, thymidine glycol, 8-OHdG and 5-hydroxymethyluracil [217]. 8-OHdG is widely used as an indicator of DNA damage because it causes a point mutation in the daughter strand of DNA [218]. Oxidative stress is associated with the activation of MMPs and the inhibition of antiproteases. Studies have shown that a higher expression of MMP-2, a gelatinase, is involved in invasion and metastasis and correlates with poor prognosis in breast cancer [219]. MMP-3 levels have been found to be increased in various subtypes of breast cancer, which stimulates the production of Rac1b (a hyperactive alternative splice form of Rac1), which in turn causes oxidative stress and DNA damage [220].

It is known that calcium-dependent hydrolases—paraoxonase (PON)—are involved in the process of antioxidant oxidation and are the products of three genes: PON1, PON2 and PON3 [221]. A number of PON1 polymorphisms lead to decreased serum PON1 activity, which contributes to increased oxidative stress. In addition, PON2 and PON3 exhibit anti-apoptotic and antioxidant effects by reducing ROS generation. Thus, paraoxonases exhibit ambiguous properties: in healthy mammary gland cells, their increased concentrations have a beneficial effect on cellular homeostasis. However, in breast cancer, these same properties may impede treatment due to their anti-apoptotic effects.

### 3.2. Proteins as Modulators of Inflammation

One of the main types of ROS is $H_2O_2$, which regulates redox signaling pathways, including NRF2 and PI3K/AKT. Today, $H_2O_2$ can be considered a "switch" of redox-dependent signal transduction pathways. The transport of $H_2O_2$ into the cell is carried out through transmembrane protein channels, aquaporins, which were discovered relatively recently. Aquaporins AQP3 and AQP5 play an important role in determining the response of tumor cells to oxidative stress (Figure 5) [222].

In breast cancer, increased AQP3 expression is associated with poor prognosis and metastasis. Mlinarić et al. studied changes in the expression of AQP3 and the antioxidant regulator NRF2 under the influence of oxidative stress at physiological levels for two weeks; the regulators Keap1 and GSK-3β were also considered [222]. The authors assessed cell viability and proliferation, as well as migration ability. Cell type has been shown to influence the effect of oxidative stress. The authors suggested that AQP3 may be part of the cellular response to oxidative stress, since in cancer cell lines, increased AQP3 expression was accompanied by increased migratory capacity and increased cell viability.

Aquaporin 5 (AQP5) has also been found to be associated with breast cancer progression [223]. Jajcanin et al., using the example of three breast cancer cell lines (MCF-7, SkBr-3 and SUM 159), examined the roles of AQP5 and NRF2 in the response to oxidative stress induced by $H_2O_2$. The results showed that this response was dependent on the stressor concentration and the expression of AQP5 and NRF2 [223].

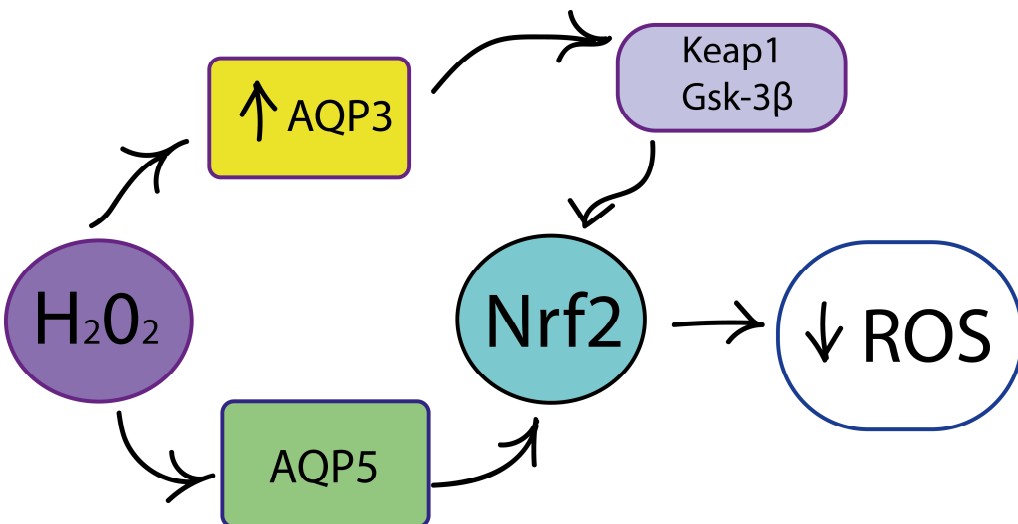

**Figure 5.** The role of transmembrane proteins aquaporins in antioxidant protection. During oxidative stress, large amounts of $H_2O_2$ enter the cell through transmembrane proteins AQP3 and AQP5, which in turn activates Nrf2, leading to a decrease in the concentration of ROS. AQP3—Aquaporin 3; AQP5—aquaporin 5; Nrf2—nuclear factor erythroid 2-related factor 2; Keap1—Kelch-like ECH-associated protein 1; Gsk-3β—glycogen synthase kinase-3 beta; ROS—reactive oxygen species.

### 3.3. Cancer Stem Cells and Oxidative Stress

Oxygen metabolism in mitochondria is a critical process for producing ATP to provide energy for cell survival and functional activity needs. However, due to electron leakage during oxidative phosphorylation and reduction of oxygen molecules, the formation of ROS can occur [224]. Mitochondrial ROS include superoxide anion, hydroxyl radical, $H_2O_2$ and singlet oxygen [224]. $H_2O_2$-mediated oxidative stress can cause premature aging in various cell types [225,226]. Zhong et al. showed that ROS levels are markedly lower in breast cancer stem cells (BCSCs) than in normal breast stem cells (NCSCs) [227]. The authors suggested that maintaining low ROS levels for CSCs may be critical to avoiding toxicity associated with oxidative stress. Loss of BCSC function caused by $H_2O_2$ treatment is associated with the induction of premature aging. A growing body of evidence suggests that BCSCs play a critical role in the emergence of drug resistance, as well as tumor progression and relapse [228]. BCSCs have been shown to have a greater DNA damage repair capacity than NCSCs [229]. Oxidative stress leads to the induction of aging. $H_2O_2$-induced loss of BCSC function and aging phenotype are associated with the activation of p53 and increased expression of p21. This is consistent with the known fact that activation of the p53/p21 signaling pathway is observed in response to DNA damage (Figure 6).

Hypoxia is an independent predictor of poor prognosis in cancer [230]. In breast cancer, hypoxia promotes vascular growth, extracellular matrix remodeling, extravasation of circulating cancer cells and increased numbers of BCSCs [231,232]. The hypoxic microenvironment promotes CSC persistence in tumors. Under hypoxic conditions, BCSC stemness is maintained through the Notch, Wnt and Hh signaling pathways, which leads to the emergence of drug resistance [233,234]. Zhou et al. found that oxidized mutant ataxia telangiectasia protein kinase (ATM), which is involved in DNA damage repair, is upregulated in hypoxic BCSCs in TNBC [235,236]. Oxidized ATM is a potential therapeutic target, as its elevated levels in breast cancer patients have been closely associated with tumor grade and progression.

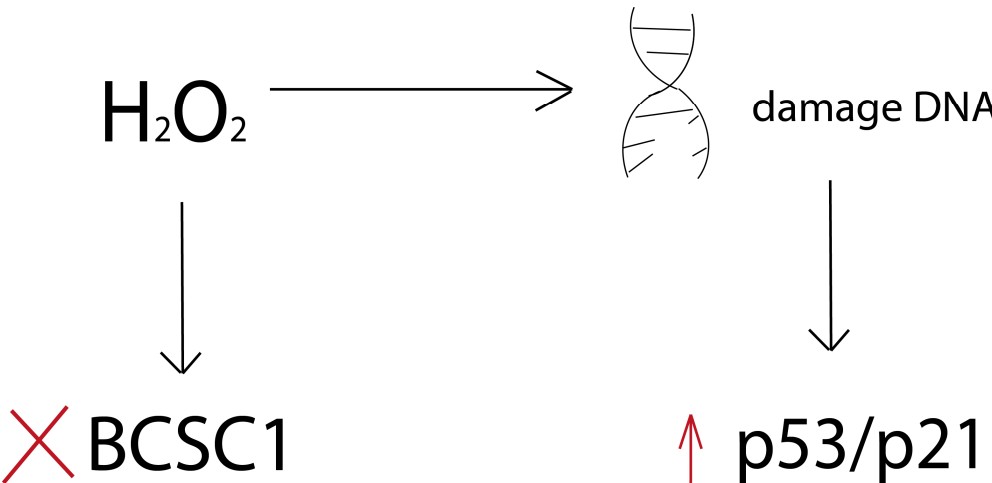

**Figure 6.** Activation p53/p21 in response to high concentrations of $H_2O_2$ and DNA damage. p53—transformation-related protein 53; p21—cyclin-dependent kinase inhibitor 1A; BCSC1—breast cancer suppressor candidate-1.

Increased expression of TGF-β1 in breast cancer is associated with increased cancer stem cell populations, local invasion, liver metastasis, and treatment resistance [237,238]. TGF-β1, like oxidative stress, may have tumor suppressor or promoter effects in cancer [239]. Various components of TGF-β1 signaling with increased expression and activation can be induced by oxidative stress; in contrast, TGF-β1 can control the cellular oxidative balance [240]. The induction of oxidative stress in breast cancer cells is usually associated with lethal damage to important intracellular structures and apoptotic cell death [241,242].

Luo et al. showed differences between mesenchymal (M-) quiescent and epithelial-like (E-) proliferative breast cancer stem cells (BCSC) in response to oxidative stress and sensitivity to glycolytic inhibitors. The main result of their study was to explain the regulation of oxidative stress through modification of the AMPK-HIF1$\alpha$ axis. This axis is responsible for the equilibrium state of BCSC and phenotypic plasticity [243]. E-BCSCs exhibit potent antioxidant responses through the activation of NRF2, thioredoxin (TXN) and GSH. Inhibition of NRF2 or synergistic blocking of TXN and GSH demonstrates limitation of the tumor's ability to further grow and metastasize. Exploiting the metabolic vulnerabilities of different BCSC states provides a new therapeutic approach targeting this critical tumor cell population.

Angiogenesis is critical for the development of blood vessels, which will subsequently supply the tumor tissue with oxygen and nutrients, thereby ensuring the vital activity of the cancer cell and promoting its progressive growth. ROS are powerful agents that penetrate endothelial cells and activate the VEGF signaling pathway, resulting in the activation of EGFR in a paracrine or autocrine manner. As a result, VEGF activates the production of endogenous ROS [244–248].

## 4. Hormone-Induced Inflammatory Response during Oxidative Stress

### 4.1. Estrogens and Oxidative Stress in Breast Cancer

An important feature of cancer cells is aerobic glycolysis, which causes a high glycolytic flux, regardless of the amount of oxygen. Due to this "Warburg effect", highly reactive glycolytic byproducts $\alpha$-oxoaldehydes such as the dicarbonyls glyoxal (GO) and methylglyoxal (MGO) are formed in significant quantities [249,250]. This process is further enhanced by oxidative stress [251], so breast cancer cells require an effective defense mechanism against aldehydes, mainly mediated by high levels of glyoxalases [250,252]. The most common and most frequently analyzed enzymes are glyoxalase I (GLO1) and glyoxalase-II (hydroxyacylglutathione hydrolase, HAGH), which use GSH as a cofactor [253]. Other glyoxalases are DJ-1, also known as PARK-7 [254] and the aldokereductase AKR7A2 [255].

In breast cancer, glyoxalase I (GLO1) levels are regulated by estrogens [256,257]. However, high concentrations of GO and MGO also cause increased formation of advanced glycation end products (AGEs), which are stable products of the Maillard reaction [258]. Glyoxalase expression is known to be important for multidrug resistance in chemotherapy [259]. Glyoxalase I has been proposed as a target for anticancer therapy [260]. Although no GLO1 inhibitor has yet been developed that has potential for use in therapy, mainly due to adverse side effects [261], exogenous application of MGO to further increase aldehyde stress has shown promising effects in a mouse model [262].

The oxidative effect of estrogen is classically associated with the production of ROS from unstable compounds, such as semiquinones, resulting from the tissue-specific conversion of estrogen to catechol estrogen metabolites (Figure 7A) [263]. Most estrogens are hydroxylated to form 2- and 4-catechol estrogens in cells, which are further oxidized to form semiquinones and quinones such as catecholestrogen-3,4-quinones, which are also the most carcinogenic form [264]. These electrophilic compounds react with DNA to form the depurinating adducts 4-OHE1(E2)-1-N3Ade and 4-OHE1(E2)-1-N7Gua. Such apurinic sites lead to mutations during DNA replication. In addition, 4-hydroxyestrogen generates free radicals through a redox cycle with the corresponding semiquinone/quinone forms, thereby causing DNA damage [265]. Mitochondria may also be involved in the generation of estrogen-associated ROS [266]. Oxidative stress caused by estrogen acts in concert with estrogen receptor-mediated signaling pathways to promote DNA damage and changes in the expression of genes responsible for cell cycle control and proliferation [267]. Estrogen can induce breast cancer cell growth through ROS-dependent regulation of epigenetic regulatory genes and epigenetic reprogramming of histone marks [268].

The process of 17b-estradiol (E2)-induced ROS production can be roughly divided into two phases depending on the response time and generation mechanisms [269]. Intracellular $Ca^{2+}$ fluctuation and ERa-dependent transcription lead to oxidative DNA damage that varies in time and space (Figure 7A). In addition, we demonstrated that DNA oxidation is required for the expression of estrogen-responsive genes. The dynamics of estrogen-induced DNA strand breaks also show a biphasic pattern, and topoisomerase-mediated DNA breaks play an important role in estrogen signaling. Lactoperoxidase is an enzyme that is produced in mammary gland cells and is involved in the one-electron oxidation of 17β-estradiol to a reactive phenoxyl radical [270]. Lactoperoxidase plays a role in breast carcinogenesis through the activation of carcinogenic aromatic amines such as benzidine, 2-aminofluorene and others, resulting in the formation of metabolites that are highly reactive and covalently bind to DNA [271].

Oxidative stress induced by estrogen may result from changes in the status of antioxidant enzymes due to estrogen receptors [272,273]. A study was conducted that showed how estrogens affect the activity of glutathione transferase, glutathione peroxidase, superoxide dismutase and catalase [274]. Estrogens and estrogen quinone metabolites can have the same damaging effects as ROS, alkylate, or damage the integrity of DNA and proteins, and also bind to estrogen receptors and activate estrogen response elements, which leads to increased levels of ROS [275]. In addition, an increased ERα/ERβ ratio in breast tumor cells demonstrates high levels of estrogen-generated ROS [276]. One of the potential causes of breast cancer is increased levels of circulating estrogens, the source of which is excess adipose tissue. In this regard, postmenopausal women are a special risk group. A direct correlation was also found between the activation of the inflammatory markers COX-2 and PGE2 and aromatase expression in obese women with breast cancer [277]. Among young women with breast cancer, on the contrary, estrogen deficiency was noted even before the start of cancer treatment. About one-sixth of them were diagnosed with TNBC [278,279].

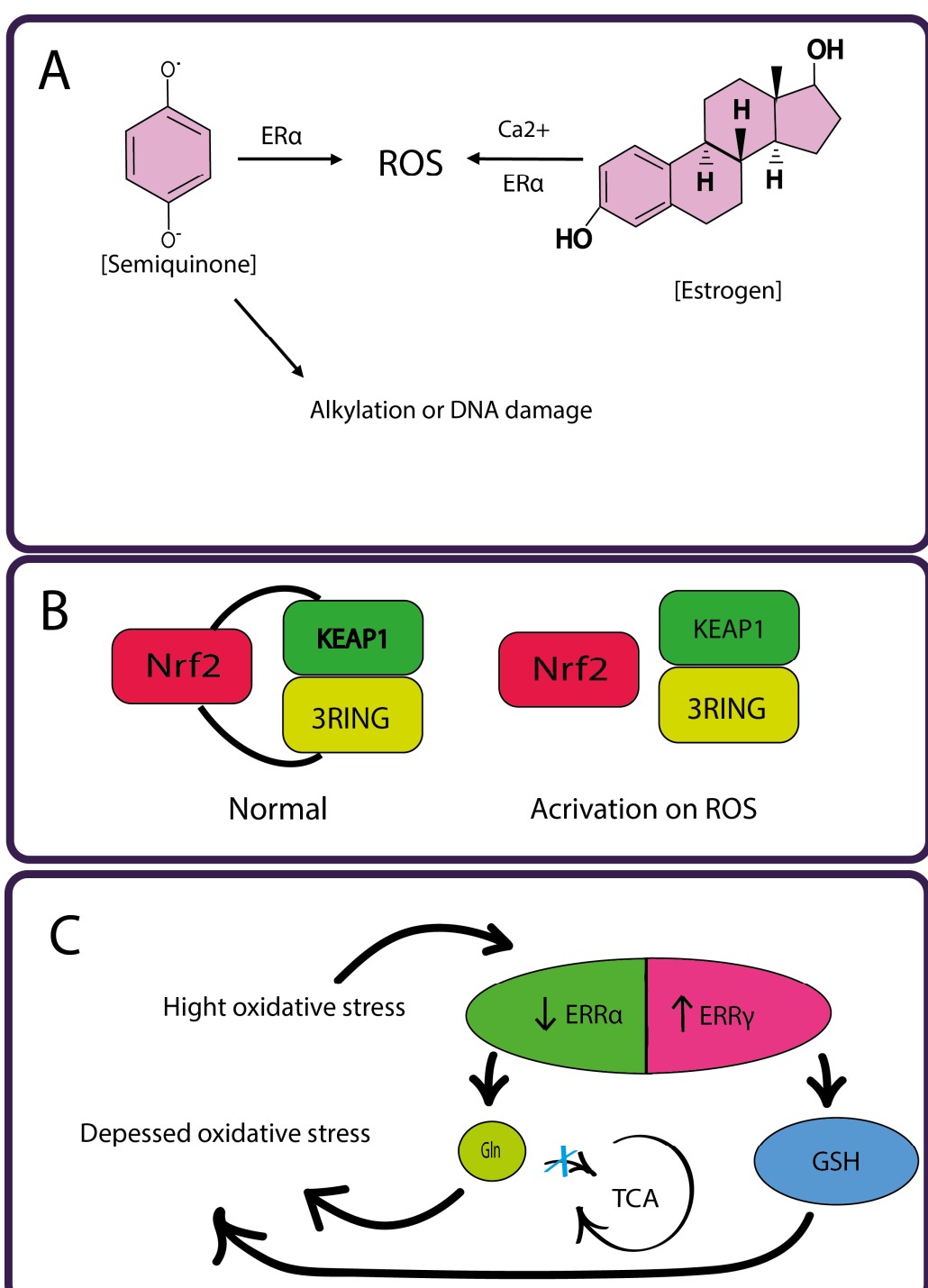

**Figure 7.** The role of estrogen in triggering oxidative stress. (**A**) Describes how estrogen metabolism products, semiquinones and 17 β-estradiol, activate production. Estradiol can also directly alkylate and damage DNA. (**B**) Under normal conditions, Nrf2 connects with the complex 3 RING and KEAP1 This state deactivates Nrf2. With an increase in the concentration of ROS inside the cell, proteolysis of the complex occurs, and 3 RING, KEAP1 and Nrf2 become active. (**C**) In the basal state, inhibition of ERRα reduces overall cellular ROS levels because decreased ERRα limits glutamine entry into the TCA cycle, whereas increased ERRγ promotes GSH production driven by glutamine. GSH—glutathion; ERRα—estrogen-related receptors α; ERRγ—estrogen-related receptors γ; Nrf2—nuclear factor erythroid 2-related factor 2; KEAP1—Kelch-like ECH-associated protein 1; 3 RING—family of RING domain-containing ubiquitin ligases; ROS—reactive oxygen species.

Estrogen promotes the development of breast cancer through estrogen receptor signaling and the production of genotoxic metabolites that cause oxidative DNA damage. Oxidative stress is counteracted by a number of genes regulated by the nuclear transcription factor erythroid-derived 2-like 2 (NRF2), encoded by the NFE2L2 gene, which initiates antioxidant responses and detoxifies xenobiotics (Figure 7B). Under normal conditions, NRF2 is ubiquitinated after direct binding to Kelch-like ECH-associated protein 1 (KEAP1) as part of the cullin 3 RING ubiquitin ligase complex and is rapidly degraded by the proteasome [280]. Under oxidative stress, NRF2 dissociates from KEAP1, stabilizes and translocates to the nucleus, where it controls the expression of numerous genes encoding enzymatic and non-enzymatic antioxidants or proteins [281]. This pathway is typically induced in response to increased levels of oxygen radicals, and increased levels of NRF2 may result from metabolic changes in transformed cells even in the absence of genetic changes directly affecting NRF2 [282]. Xie et al. studied a series of novel direct Keap1-Nrf2 PPI inhibitors and their role in increasing the availability of Nrf2 for antioxidant activity and attenuating estrogen-mediated responses in breast cancer [283]. It has been shown that the NRF2 gene signature can potentially be used to identify patients who are not likely to experience metastatic recurrence [284], as cancer cells experience high levels of oxidative stress during metastasis [285].

From a cellular homeostasis perspective, autophagy is a critical response to oxidative stress. Like autophagy, ROS also have a dual nature in carcinogenesis [286,287]. Various treatments can activate ROS-induced autophagy, which in turn may lead to drug resistance or the induction of apoptosis [288]. The autophagy pathway is critically modulated by ROS, while ROS help stimulate autophagy. However, the effects of ROS and autophagy differ depending on the stage of tumor development. Thus, in the future, cancer treatment may include antioxidants or autophagy inhibitors to reduce ROS-induced cytoprotective autophagy during treatment [289,290].

The estrogen-bound nuclear receptors ERRα and ERRγ are redox sensors that play a key role in the control of energy metabolism in both normal and cancer cells [291,292]. ERRs regulate the expression of genes involved in glycolysis, fatty acid oxidation and mitochondrial biogenesis, as well as functions including oxidative phosphorylation and electron transport [293]. Both ERRα and ERRγ are involved in the regulation of ROS signaling (Figure 7C) [294]. In the basal state, inhibition of ERRα reduces overall cellular ROS levels because decreased ERRα limits glutamine entry into the TCA cycle, whereas increased ERRγ promotes glutathione production driven by glutamine. In contrast, under oxidative stress, cells require the activation of both receptors to initiate an adaptive antioxidant response to stress. In this case, increased expression and activity of ERRγ may serve as a sign of oxidative stress and could be used therapeutically to sensitize breast cancer cells to chemotherapy [291].

## 4.2. Stress Hormones and Breast Cancer

Psychological stress is known to cause an increase in circulating levels of stress hormones, including cortisol [295]. A number of epidemiological studies have shown the relationship of negative psychosocial factors with an increase in the incidence of breast cancer, worse survival [296] and metastasis [297]. In particular, glucocorticoid receptor (GR)-mediated glucocorticoid signaling promotes tumorigenesis and drug resistance in TNBC [298], and increased GR expression in breast tumors correlates with decreased survival [299]. In turn, GR antagonism induces apoptosis and, in combination with conventional chemotherapy, reduces tumor size in TNBC models (Figure 8A) [300]. Glucocorticoids are known to have non-genomic effects on inducible nitric oxide synthase (iNOS) and enhance nitric oxide signaling in breast cancer cells [301]. Thus, iNOS can be considered a potential therapeutic target in breast cancer [302], since iNOS expression is positively correlated with tumor grade, stage and metastasis [303,304]. Flaherty et al. also showed that cortisol induces the expression of key angiogenesis-related genes and protumorigenic immunomodulation [305]. Thus, the glucocorticoid cortisol induces the expression of key

genes associated with angiogenesis, as well as protumorigenic immunomodulation. These results demonstrate the detrimental involvement of NOS in stress hormone signaling and the potential future benefits of NOS inhibition in patients with severe stress (Figure 8B).

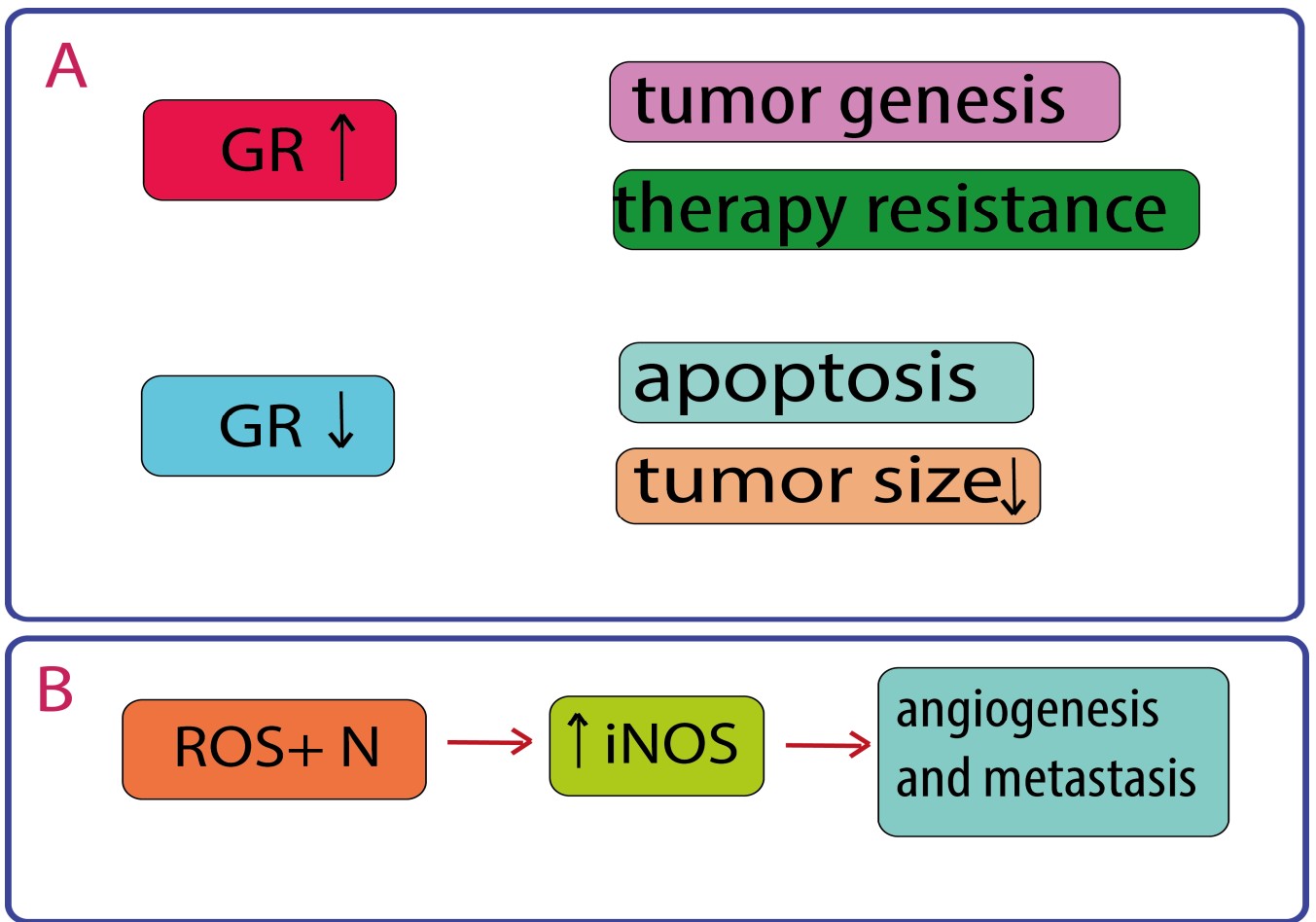

**Figure 8.** The role of stress hormones in breast cancer. (**A**) Increased activity of glucocorticoid receptors leads to potentiation of tumorigenesis and resistance to drug therapy. A decrease in the activity of glucocorticoid receptors leads to cell apoptosis and a decrease in tumor size. (**B**) ROS interact with nitrogen to form induced nitric oxide synthase. This leads to the activation of angiogenase in the tumor and its metastasis. GR—glucocorticoid receptor; iNOS—inducible nitric oxide synthase; ROS—reactive oxygen species. Up arrow means increased activity; down arrow means decreased activity.

Mental stress in breast cancer typically manifests as anxiety, depression, nervousness and insomnia, all of which cause stress reactions [306]. The continuous release of neurotransmitters from the neuroendocrine system may have a profound impact on the occurrence and prognosis of breast cancer [307]. The role of neurotransmitters in breast cancer is complex and varied. Bae et al. analyzed the relationship between chronic stress and breast cancer, in particular the molecular mechanism by which chronic stress promotes the development of breast cancer [308].

## 5. Epigenetic and Genetic Regulation of Oxidative Stress in Breast Cancer
### 5.1. BRCA1 and Oxidative Stress

High levels of oxidative stress are associated with breast cancer aggressiveness [309–311]. The breast cancer early-onset gene 1 (BRCA1) functions as a guardian of genome stability and has multiple effects on key cellular processes involved in DNA damage repair, transcriptional

regulation, ubiquitination and cell cycle control [312]. Some results suggest that BRCA1 may generally protect against oxidative stress. Under basal conditions, the inhibition of BRCA1 expression was observed with increasing concentrations of $H_2O_2$ as an inducer of oxidative stress. In contrast, the forced overexpression of BRCA1 in vitro activated NADPH oxidase Nox1 and its regulatory subunit p47phox and inhibited $H_2O_2$-induced ROS production [313]. Overexpression of BRCA1 in breast cancer cell lines increases the expression of several genes involved in antioxidant responses, such as glutathione S-transferase, by enhancing the antioxidant response of the transcription factor NRF-2, which in turn reduces ROS levels and confers resistance to $H_2O_2$ exposure [314,315]. BRCA1 is also involved in the regulation of other DNA repair pathways, including base excision repair of small oxidative DNA lesions, both directly through transcriptional activation and indirectly by acting as a scaffold [316].

The main DNA base damage caused by ROS is 8-oxoguanine (8-oxoG), which is repaired by oxoguanine glycosylase 1 (OGG1) [317]. BRCA1 transcriptionally activates OGG1 upon generation of 8-oxoG for its repair [316,318]. Inactivation of BRCA1 causes high levels of oxidative stress with increased production of superoxide anions and $H_2O_2$ and increases cell sensitivity to oxidative stress, reducing cell viability [319]. Thus, BRCA1 may function as a natural endogenous antioxidant. Martinez-Outschoorn et al. showed that loss of BRCA1 function in breast cancer cells and stromal fibroblasts leads to the formation of $H_2O_2$, which in turn promotes a reactive glycolytic stroma with increased MCT4 and decreased caveolin-1 (Cav-1) expression [309]. Importantly, these metabolic changes in BRCA1 deficiency can be reversed by the use of antioxidants, which cause cancer cell death. For example, a number of antioxidants (N-acetyl cysteine, resveratrol and selenium) are promising as prophylaxis for patients with BRCA1 mutations [320–322]. At the same time, the authors suggest that Cav-1 and MCT4 immunostaining can be used as biomarkers to monitor the response to antioxidant therapy [309].

It has been suggested that estrogen may play an important role in the development of breast cancer in carriers of BRCA1 mutations, which is especially pronounced during periods of active cell proliferation and differentiation in the mammary gland. These periods include puberty and pregnancy, when BRCA1 expression and ovarian hormone production typically increase [323]. Therefore, women with reduced BRCA1 protein expression due to inheritance of a BRCA1 mutation may be particularly susceptible to the carcinogenic effects of hormonal exposure. In particular, there may be persistent DNA damage that can result from persistent ER-dependent transcriptional activation, which is restored by BRCA1 in normal mammary epithelium but is exacerbated by subsequent loss of BRCA1 [324].

Several studies have reported a controversial role for NO in the contextual activation of pathways involving BRCA1 or in the suppression of BRCA1 expression [325,326].

Checkpoint kinase (2CHEK2) encodes a protein that is involved in maintaining genome stability and regulates the processes of cell division and DNA repair. Gene activation, followed by protein synthesis, occurs in response to damage to the DNA molecule, blocking the cell cycle in G1 or triggering the process of apoptosis (programmed cell death). Thus, CHEK2 is a suppressor gene of moderate effect and plays an important role in protecting cells from inflammation of any origin and oxidative stress. In addition, it is an important predictor of predisposition to the development of breast cancer. Oxidative stress causes the activation of the KEAP1-NRF2 and ATM-CHEK2-p53 pathways. It has been shown that in human breast tissue, high levels of ROS are activated and transported by basoluminal precursors, which are direct precursors of basal-like breast cancer [327,328].

Partner and localizer of BRCA2, FANCN (PALB2), like CHEK2, acts as a tumor suppressor, takes part in the repair of damaged DNA and promotes the expression of antioxidant genes. The PALB2 gene is part of the BRCA1-PALB2-BRCA2 functional chain. In addition, PALB2 promotes the accumulation of one of the main antioxidant transcription factors, NRF2. This results in a reduction in oxidative stress due to a strong antioxidant response [329].

## 5.2. p53 and Oxidative Stress

Oxidative stress is known to induce activation of the TP53 gene and consequently the production of the tumor suppressor protein p53 to prevent further DNA damage [330].

It has been shown that the protumorigenic or antitumor effect of an increase in nitric oxide (NO) levels depends on the status of the p53 tumor suppressor gene [331]. Increased NO levels ensure stabilization and accumulation of p53, but activation of p53 suppresses NOS2 activity in a negative feedback manner [332].

Transcriptional targets of the p53 tumor suppressor gene have also been shown to include antioxidant enzymes and the RRM2B genes, which regulate the codification of the ribonucleotide reductase subunit, preventing DNA dysfunction in mitochondria. This process involves the ATM gene, which stabilizes mitochondrial DNA by regulating ribonucleotide reductase [333]. Sablina et al. reported that low concentrations of p53 under conditions of low cellular stress promote the expression of antioxidant genes. The oxidative function of p53 is to increase the expression of genes that contribute to an increase in ROS and induction of apoptosis [334].

P53 expression is closely related to iron metabolism. Excess iron leads to decreased expression of p53 [335], whereas iron depletion leads to accumulation of p53 [336]. In addition, direct binding of heme to the p53 protein inhibits p53 transcriptional activity and possibly promotes p53 degradation [335]. Interestingly, p53 can promote and inhibit ferroptosis depending on the context [337]. For example, p53 is able to inhibit ferroptosis through p21, a major target of p53 that inhibits glutathione degradation [338]. P53 may promote ferroptosis by directly inhibiting the expression of SLC7A11 (a component of the cystine/glutamate antiporter) [339]. SLC7A11 is overexpressed in human tumors, and the p53-mediated regulation of SLC7A11 and ferroptosis modulates cancer cell survival. Reducing SLC7A11 levels under conditions of basal p53 regulation or low stress provides another level of protection against tumorigenesis by lowering the stress tolerance threshold for ROS [340], and allows programmed cell death to be activated to avoid potential harm from genetic instability [341].

## 5.3. Epigenetic Regulation of Oxidative Stress in Breast Cancer

A new but rapidly developing area of research is mitoepigenetics, or epigenetic regulation of mitochondrial DNA (mtDNA) [342]. The structure of the inner mitochondrial membrane attracts special attention from scientists, since it is where a large amount of ROS is concentrated [343]. Mitochondria play a very important role in the regulation of intracellular processes. These include regulation of aerobic respiration, proliferation and survival of cells, their apoptosis, as well as the synthesis of nucleic acids and oxidative stress. Dysregulation of cascade responses associated with ROS production has been associated with cancer progression [344]. It is known that defects in the mtDNA repair system and mitochondrial nucleotide protection in cancer cells are ROS-mediated by oxidative stress. Defects in the collective mitochondrial genome have been shown to contribute to the initiation and progression of breast cancer [345,346]. Mutational damage to mtDNA, its excessive proliferation and deletions introduce changes into the nuclear epigenetic landscape [347]. The ability of cancer cells to survive is maintained by unbalanced mitoepigenetics and dysregulation of oxidative phosphorylation (OXPHOS) [348]. A number of therapeutic targets (biguanides, OXPHOS inhibitors, vitamin E analogues and the antibiotic bedaquiline) have been proposed for future clinical trials in patients with breast cancer [349–351]. Mutations and epigenetic modifications of mtDNA can be used as potential markers for the early diagnosis and targeted therapy of breast cancer [352].

The redox protein p66ShcA is known to be differentially expressed in cancer [353]. In a steady state, p66ShcA is located in the cytoplasm and acts as an adapter protein [354,355]; however, after activation in the inner membrane space of mitochondria, it enhances the transfer of electrons from cytochrome C to oxygen with the formation of mitochondrial ROS (Figure 9) [356,357].

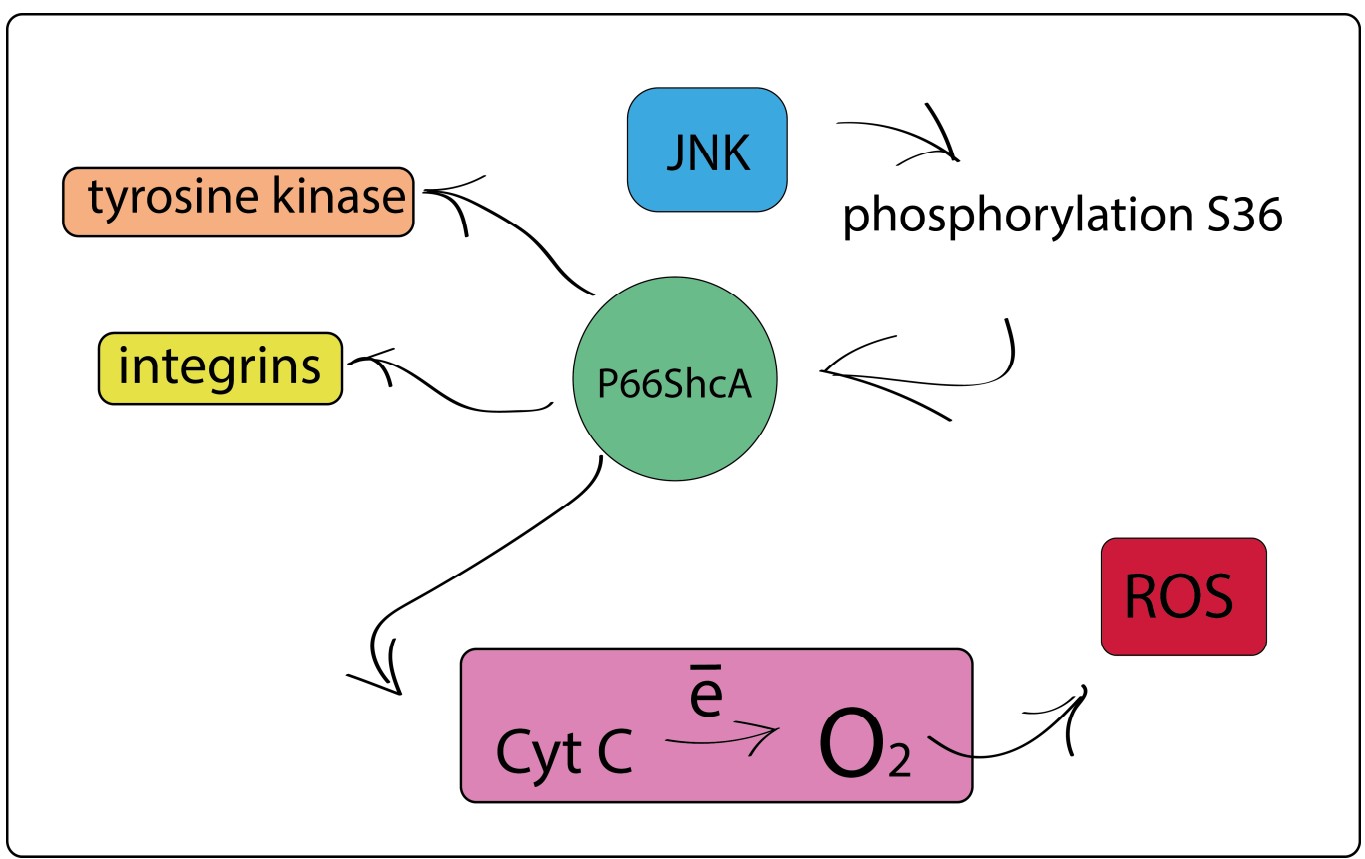

**Figure 9.** The role of redox protein p66ShcA in the formation of ROS and potentiation of oxidative stress through tyrosine kinase and integrin proteins. Protein activation of p66ShcA occurs because JNK phosphorylate a unique serine residue (S36). This protein activates mitochondrial cytochrome C and transfers it to oxygen with a subsequent increase in ROS production. The redox protein p66ShcA also activates the action of tyrosine kinase and affects integrin proteins. Cyt C—cytochrome C; S36—serine residue; JNK—c-Jun N-terminal kinases; ROS—reactive oxygen species.

The association of elevated p66ShcA levels with breast cancer prognosis is controversial, with both good outcomes [358] and increased recurrences [359] reported. This depends on the degree of increase in oxidative stress under the influence of p66ShcA. Thus, with a significant increase in ROS, they destroy the membrane potential of mitochondria, which leads to the release of cytochrome C and apoptosis [357]. This is confirmed by the p66ShcA-induced enhancement of the cytotoxic effect of chemotherapy [360]. In contrast, moderately elevated oxidative stress enhances the metastatic potential of breast cancer [361–363]. Lewis et al. showed that p66ShcA increases the metabolic flexibility of estrogen receptor (ER)-negative breast cancer, independent of HER2 status [364]. In hormone-sensitive tumors, elevated levels of ROS provide them with a growth advantage by inactivating redox-sensitive protein tyrosine phosphatases to enhance receptor tyrosine kinase signaling [365].

Another epigenetic target in breast cancer is the G9a enzyme, which catalyzes histone methylation, thereby influencing gene expression [366]. Modifications of histones and gene expression in breast tumors require histone methyltransferase G9a, while tumor recurrence or suppression may be due to the genetic or pharmacological inhibition of G9a [367,368]. Thus, UNC0642 (a selective small-molecule inhibitor of G9a) effectively reduces genomic H3K9 methylation and has potent anticancer activity [368]. Singh et al. showed in three breast cancer cell lines (MDA-MB-231, T47D and MCF-7) that TXNIP activity was increased after treatment with UNC0642 [369]. TXNIP is an mRNA that transmits redox signaling by inhibiting thioredoxin activity [370]. The authors suggest that increased ROS production

together with TXNIP contributes to mitochondrial damage and activation of apoptotic pathways [370]. Low levels of ROS promote cell proliferation, while high levels cause excessive oxidative damage and lead to cell death [371,372].

Activation of oncogenes in breast cancer can lead to the simultaneous induction of oxidative stress and endoplasmic reticulum stress, which triggers the integrated stress response (ISR) [373]. In this regard, the long noncoding RNA UBA6-AS1 was identified as being upregulated by amino acid deficiency and stress, such as arginine or glutamine deprivation or treatment with endoplasmic reticulum stress inducers in TNBC. Induction of UBA6-AS1 promotes the survival of TNBC cells under metabolic stress, suggesting a regulatory role for UBA6-AS1 in response to intratumoral metabolic stress during tumor progression [373].

## 6. Discussion

Thus, pathological oxidative stress is one of the fundamental pathophysiological mechanisms of carcinogenesis, uniting all body systems. The map of connections between participants influencing the increase in ROS content consists of four large sections: biochemical processes associated with the metabolism of amino acids, metals and glycolysis; description of individual components of inflammation; hormonal-induced oxidative stress and its main regulators; genetic and epigenetic regulation of oxidative processes inducing carcinogenesis (Figure 10).

Until now, the question remains whether oxidative stress is one of the causes of breast cancer or whether it is a consequence of the development of breast carcinoma. It is known that tumor cells produce excessive amounts of ROS due to the dysregulation of NADPH oxidase [374]. Breast cancer cells are characterized by the overexpression of thymidine phosphorylase, which catalyzes the breakdown of thymidine to thymine and 2-deoxyribose-1-phosphate [375,376]. 2-Deoxyribose-1-phosphate is involved in the protein glycation reaction, which is accompanied by the generation of ROS. Finally, a breast cancer-specific mechanism for ROS generation has been proposed, called estrogen hormone metabolism by lactoperoxidase [271,272]. Macrophage infiltration of the tumor makes a certain contribution to the induction of oxidative stress in breast cancer. Macrophages produce not only ROS but also tumor necrosis factor $\alpha$, which is known to induce intracellular oxidative stress. It has been suggested that tumor angiogenesis is accompanied by alternating periods of hypoxia and reperfusion, which also leads to excessive generation of ROS [62]. The generation of ROS not only causes direct damage to biopolymers but also promotes the activation of intracellular signaling pathways, which also lead to overproduction of ROS [6,7,11–14]. According to Mencalha et al., long-term mitochondrial dysfunction is the main mechanism that leads to oxidative modification of proteins and lipid peroxidation, as well as a high probability of direct damage to DNA and the repair system [377]. All this together leads to malignant transformation of breast cells, creating a localized tumor mass characteristic of the early stages of the disease. The constant formation of ROS and oxidative stress products promotes tumor metastasis, affecting the aggressiveness of breast cancer. Oxidative stress may also be involved in the induction of chemically resistant breast cancer and may be considered a hypothesis to explain disease recurrence and secondary tumor development. Oxidative stress may also be a factor influencing the prognosis of the disease, since chemotherapy drugs can affect the redox status of breast cancer patients in various ways [378].

Another direction of research into oxidative stress in breast cancer patients is to study the consequences of its development for tumor cells [379]. On the one hand, ROS induce cell entry into apoptosis; on the other hand, persistent oxidative stress leads to the development of resistance to apoptosis. Persistent oxidative stress in breast carcinoma cells can induce preferential selection of cells with an apoptosis-resistant phenotype in which the p53 gene is turned off [380]. This has important implications for breast cancer treatment [381]. The antitumor efficacy of radiotherapy, photodynamic therapy and chemotherapy is based on their ability to generate ROS and induce apoptosis of tumor cells [371,382]. However, per-

sistent oxidative stress in breast cancer tumor cells may cause treatment resistance [383,384]. Oxidative stress can induce angiogenesis in breast carcinoma [244]. The growth of blood vessels in the breast tumor microenvironment increases the risk of blood-related metastasis. Oxygen free radicals can also increase tumor cell migration [385]. Modeling studies have suggested that the mechanism of this process is associated with the activation of P38 MAPK, which in turn increases the phosphorylation of heat shock protein-27, which promotes the migration of breast cancer cells [385]. According to other researchers, oxidative stress in breast tumor cells promotes metastasis through cytoskeletal reorganization [386]. It has been suggested that oxygen radicals can increase the resistance of breast carcinoma cells to chemotherapy due to the expression in tumor cells of P-glycoprotein, which is an efflux pump that determines multidrug resistance (the multidrug-resistance efflux pump) [387]. It has been established that low and moderate concentrations of ROS can contribute to tumor formation, either by acting as signaling molecules or by inducing mutations in genomic DNA. At the same time, high levels of ROS contribute to both cell death and deep damage to cellular structures. In the early stages of tumor development, some authors consider this phenomenon positive. However, there is evidence that high levels of ROS promote the detachment of cells from the cellular matrix. In this regard, finding ways to increase oxidative stress is a strategic direction that can destroy cancer cells. In this regard, the direction associated with the induction of oxidative stress in the endoplasmic reticulum seems promising. With the development of oxidative stress in the endoplasmic reticulum, the conditions for the normal folding of synthesized proteins are disrupted, which leads to the accumulation of the latter and the activation of apoptosis. Such stress inducers are currently considered promising anticancer drugs [282].

Of great interest is the formulated concept of the role of oxidative stress in the microenvironment of breast carcinoma [282]. According to modern concepts, breast carcinoma includes both mutated somatic cells and a microenvironmental system, which includes fibroblasts, adipocytes, immune and endothelial cells. Oxidative stress plays an important role in the initiation and maintenance of breast cancer progression. ROS are generated in various metabolic pathways, including the mitochondrial electron transport chain, activation of NADPH oxidases (NOX), etc. Model studies have established that in breast cancer, ROS are formed both in tumor cells and in the cancer stroma and are capable of inducing cell migration. ROS, especially $H_2O_2$, are also able to stimulate the growth of normal fibroblasts into myofibroblasts, which in turn are able to generate large amounts of $H_2O_2$, increasing oxidative stress in the microenvironment [388]. It has been established that in breast cancer, up to 80% of fibroblasts demonstrate an activated phenotype. Myofibroblasts secrete collagen type I, which increases breast density, which in turn promotes breast cancer formation and metastasis. In addition, there is an increase in the secretion of various growth factors (TGFβ, IGF, PDGF, etc.), which have a number of effects, including stimulating the formation of ROS through NOX. In other words, a large amount of ROS is generated in the tumor microenvironment, which initiates or promotes the progression of breast cancer. Under conditions of mitoptosis, as well as with enhanced antioxidant protection, ROS act as factors limiting tumorigenesis. Thus, an increased level of ROS production activates endogenous antioxidant defense mechanisms, such as the redox-dependent antioxidant response element Keap1/Nrf2/ARE system [280–285] and autophagy [286–290]. Inhibiting oxidative stress in the microenvironment using antioxidants is a very attractive approach for breast cancer therapy. However, adding a cocktail of antioxidants turned out to be ineffective in preventing cancer, and in some cases, on the contrary, provoked its development. This clearly highlights the need for a more detailed understanding of how oxidative stress contributes to cancer development. The mechanism of action of a number of chemotherapy drugs is associated with increased ROS generation. It is proposed to classify chemotherapy drugs into two categories: drugs with direct and indirect effects on ROS metabolism. For example, drugs such as platinum coordination complexes and anthracyclines promote the generation of extremely high ROS levels, which can trigger the apoptosis of tumor cells. The mechanism of action of taxanes is associated with their

influence on the release of cytochrome C, with increased generation of ROS in mitochondria and induction of apoptosis [5]. This made it possible to formulate the concept of modulators of oxidative stress in breast cancer as one of the directions of the antitumor strategy. In particular, melatonin and polyphenol are considered such modulators [389]. Other researchers have substantiated strategies for breast cancer therapy and prevention that target the redox system [389,390]. This is due to the fact that components of the redox system, for example, GSH and thioredoxin, are involved in protecting tumor cells from apoptosis and in the formation of multidrug resistance. Based on this, one of the directions is associated with the development of drugs that block the synthesis of GSH and thioredoxin in tumor cells [41]. Today, active research is being conducted on the possibility of regulating the redox balance of tumor cells: antioxidants with targeted action are being synthesized, specific inhibitors of the enzymatic mechanisms of ROS production and antioxidant protection are being developed, and combinations of different antioxidants are being used [391]. An interesting direction is the creation of hybrid antioxidants that act on different ROS. In MCF-7 breast adenocarcinoma cells, one of these compounds, sodium 3-(3′-tert-butyl-4′-hydroxyphenyl)-propylthiosulfonate (TC-13), induced apoptosis, and the resistance of tumor cells to doxorubicin decreased [392].

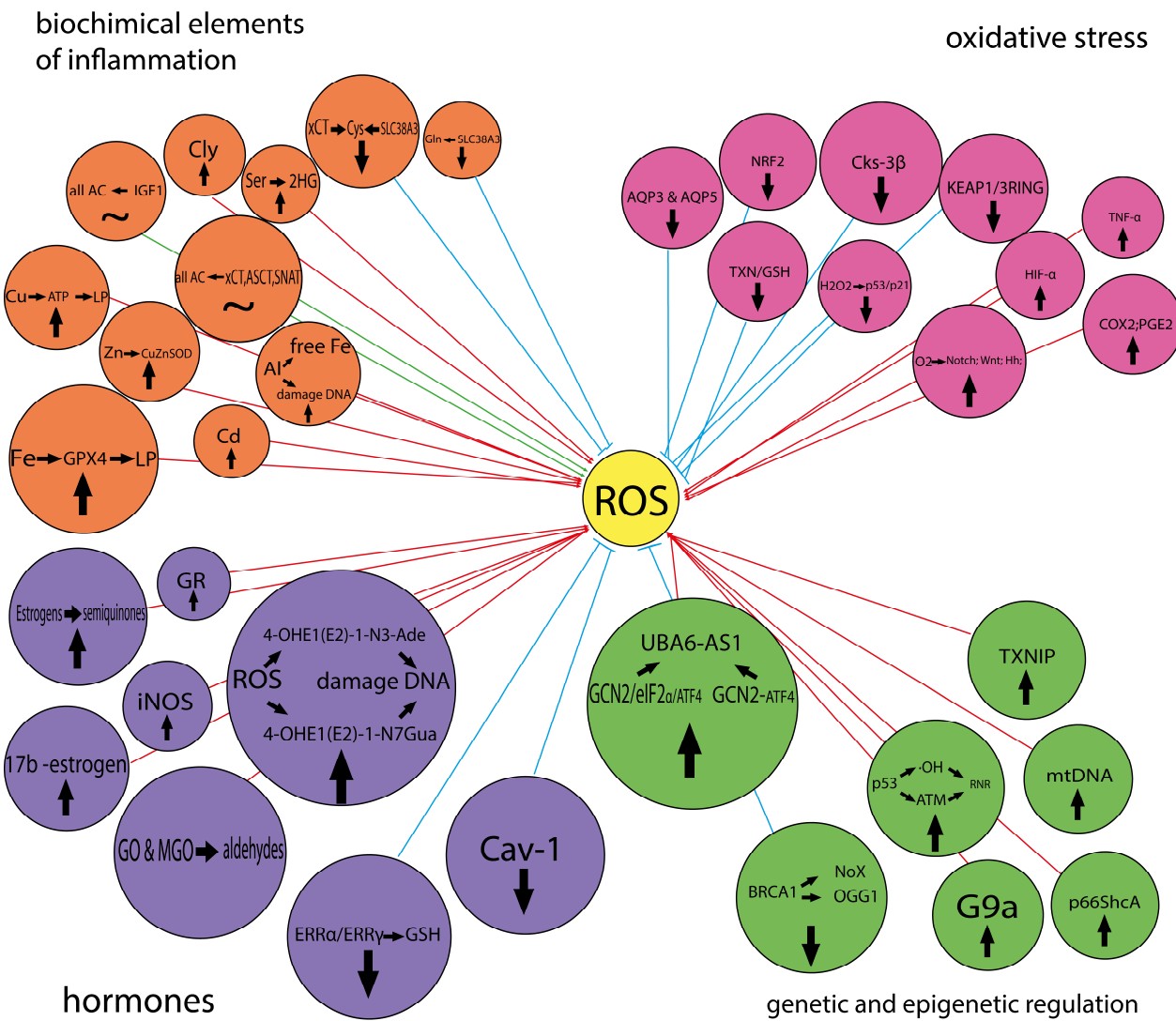

**Figure 10.** Biochemical map of ROS production. Arrows in circles that point down indicate a decrease in ROS production. Arrows pointing up indicate an increase in ROS production. The tilde denotes an ambiguous effect on ROS production. The arrows inside the circles describe the main participants in

the metabolic pathway, leading to either an increase or decrease in ROS production. Arrows pointing away from the circles: arrows that induce ROS production are indicated in red; arrows that inhibit the production of ROS are indicated in blue; arrows that have ambivalent properties are indicated in green (on the one hand they induce, on the other hand they inhibit. Biochemical elements of inflammation: AC—amino acid; IGF1—insulin-like growth factor 1; Gly—glycine; Ser—serine; 2HG—2-hydroxyglutarate; xCT—cystine/glutamate antiporter xCT; ASCT2—novel alanine serine cysteine transporter 2; SNAT—serotonin N-acetyltransferases; Cys—cysteine; SLC38A3—solute carrier family 38 member 3; Gln—glutamine; ATP—adenosine triphosphate; LP—lipid peroxidation; CuZnSOD—copper–zinc superoxide dismutase; GPX4—glutathione peroxidase 4. Oxidative stress: AQP3—aquaporin 3; AQP 5—aquaporin 5; Nrf2—the nuclear factor erythroid 2–related factor 2; TNX/GSH—thioredoxin/glutathione; Cks-3β—glycogen synthase kinase-3 beta; $H_2O_2$—hydrogen peroxide; p53—a tumor suppressor gene; p21—a cyclin-dependent kinase inhibitor; KEAP1—kelch-like ECH-associated protein 1; 3RING—family of RING domain-containing ubiquitin ligases; HIFα—hypoxia-inducible factor α; TNFα—tumor necrosis factor-α; Notch, Wnt, Hh—signaling pathways; COX2—cyclooxygenase 2; PGE2—prostaglandin E2. Hormones: RG—glucocorticoid receptor; iNOX—inducible nitric oxide synthase; GO—dicarbonyls glyoxal; MGO—methylglyoxal; 4—OHE1(E2)-1-N3-Ade and 4—OHE1(E2)-1-N7Glu—estrogen metabolites; ERRα/ERRγ—estrogen-related receptor α/estrogen-related receptor γ; Cav-1—caveolin 1. Genetic and epigenetic regulation: UBA6-AS1—long non-coding RNA UBA6-AS1; GCN2/eIF2a/ATF4—general control nonderepressible 2/eukaryotic initiation factor 2/activating transcription factor 4; GCN2-ATF4—general control nonderepressible 2-activating transcription factor 4; BRCA1—breast cancer antigen 1; NoX—NADPH oxidases; OGG1—oxoguanine glycosylase; ·OH—hydroxyl radical; ATM—ataxia telangiectasia protein kinase; RNR—ribonucleotide reductase; TXNIP—thioredoxin-interacting protein; mtDNA—mitochondrial DNA; G9a—euchromatic histone-lysine N-methyltransferase 2 (EHMT2), also known as G9a; p66ShcA—a pro-apoptotic protein that regulates oxidative stress and induces the mitochondrial apoptosis pathway by means of redox activity.

The limitations of the study are primarily due to the lack of information about the metabolic behavior of the different molecular subtypes of breast cancer. A large number of studies examine individual metabolic processes, which does not allow for a more complete and systematic description of the causes and consequences of oxidative stress and its impact on the occurrence and progression of breast cancer.

## 7. Conclusions

The role of ROS in carcinogenesis appears to be much deeper and broader than was considered in the past. The uniqueness of this review lies in the fact that we demonstrated that the process of ROS formation could have a wide variety of triggers and a large cascade of reactions. The key observation of this review is that it is possible to identify both new key players in oxidative stress and to find common, to date, not yet fully understood relationships between participants in inflammatory reactions from different regulatory systems leading to the appearance of mammary glands, contributing to its progression. We tried to provide the most complete information on the participation of ROS during oxidative stress in breast cancer cells at the biochemical level, in terms of enzymatic reactions, amino acid metabolism and metal metabolism. We examined the main possible ways to maintain the stemness of cancer cells and provided the most complete list of proteins that take an active part in the modulation of inflammatory reactions, which trigger an aggressive environment and potentiate carcinogenesis. Hormonal imbalance of estrogen and stress hormones has been described as one of the potential causes of breast tumors with a detailed description of specific signaling pathways. An important component of the appearance of breast neoplasia is a disturbance in epigenetic regulation. This is a relatively new area of research that will allow us to look at the main points of the regulation of cancer cell formation from a new angle. The information we have systematized is valuable in the field of oncology as it forms a deeper understanding of the nature of the life activity of breast cancer cells and the role of oxidative stress. The mechanisms that trigger carcinogenesis

are described in sufficient detail, and up-to-date information on possible new therapeutic targets is provided.

The further goal of the study will be a more systematic and detailed description of the not only the relationships between certain components that take part in and trigger the production of ROS, but also their internal relationships with each other.

**Author Contributions:** Conceptualization, L.V.B. and E.I.D.; methodology, L.V.B.; software, E.I.D.; validation, L.V.B. and E.I.D.; formal analysis, L.V.B.; investigation, L.V.B.; resources, L.V.B.; data curation, E.I.D.; writing—original draft preparation, E.I.D.; writing—review and editing, L.V.B.; visualization, E.I.D.; supervision, L.V.B.; project administration, L.V.B. All authors have read and agreed to the published version of the manuscript.

**Funding:** This research was funded by the Russian Science Foundation, grant number 23-15-00188, https://rscf.ru/project/23-15-00188/ (accessed on 10 May 2024).

**Conflicts of Interest:** The authors declare no conflict of interest.

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
