# Peer review of "Oxidative Stress in Breast Cancer: A Biochemical Map of Reactive Oxygen Species Production"

_cimb, doi:10.3390/cimb46050282_

Round 1

Reviewer 1 Report (New Reviewer)

Comments and Suggestions for Authors

Revision Manuscript ID cimb-2974826

The authors of the manuscript entitled Oxidative stress in breast cancer: А biochemical map of reactive oxygen species production.

This review contains valuable information about the molecular characteristics of breast cancer related to oxidative stress. It is an interesting read for anyone interested in this topic.

The manuscript, with many figures and tables, seems well organized and comprehensively described.

Minor revision is required.

It's recommended that the references be updated and reduced in number. An excess of 395 references can be overwhelming for a review, and more recent references would enhance the manuscript's currency.

Usually, the discussion part is not included in a review manuscript. However, you could resume some essential concepts in the introduction and the conclusions.

English should be corrected in some places for better understanding.

Comments on the Quality of English Language

The English should be corrected in the manuscript using software such as Grammarly for better understanding.

Author Response

Reviewer 2 Report (New Reviewer)

Comments and Suggestions for Authors

The authors of this review aim to summarize information on the metabolic characteristics of breast cancer closely linked to oxidative stress. It is evident that the generation of reactive oxygen species (ROS) can be triggered by a wide range of factors, leading to an extensive cascade of reactions. The systematic compilation of information enables us to comprehensively analyze the role of oxidative stress in the onset and progression of breast cancer, as well as evaluate the contribution of each of these aspects. The mechanisms initiating carcinogenesis are detailed, and an updated view on potential new therapeutic avenues is provided. This review could potentially be of great significance in its field. It is extensive yet written in a clear and concise manner. Acceptance is suggested after the authors clarify the following points:

-       The following statement is confusing: "Glucose deprivation causes a switch from aerobic glycolysis to oxidative phosphorylation, and ROS production by the mitochondrial respiratory chain promotes oxidative stress and apoptosis [28]." Aerobic glycolysis is closely associated with oxidative phosphorylation. Why would there be a shift from aerobic glycolysis to oxidative phosphorylation? Could it be referring to anaerobic glycolysis instead? Throughout the article, aerobic glycolysis and oxidative phosphorylation are mentioned as distinct processes. If the intention is to describe a scenario where glycolysis occurs without pyruvate entering the Krebs cycle and oxidative phosphorylation, even in the presence of oxygen, this needs to be clarified.

-       Figure 1: Please provide an explanation of the figure first, followed by the meaning of the abbreviations. Why is GLUT-1 illustrated? Which tissue are we referring to? Please specify.

-        The authors mention: "As a protective response against ROS overproduction, cancer cells have acquired changes in metabolic pathways involved in maintaining redox homeostasis. These pathways include the pentose phosphate pathway or glutamine uptake, which support NADPH and GSH synthesis, respectively." However, what specific modifications are made?

-        It is mentioned that well-oxygenated cancer cells preferentially utilize lactate instead of glucose. But how does this work? Do they engage in gluconeogenesis? Or why do they use lactate to generate energy?

-       Figure 3: Please provide an explanation of the figure first, followed by the meaning of the abbreviations. The authors may consider including the enzymes involved in glutathione synthesis.

-       ¿ What factors determine the action of copper as both an anticancer agent and a pro-carcinogen? This concept is addressed in section 2.3; however, it is poorly explained and confusing.

-       Figure 4: Please begin with an explanation of the figure, followed by the meanings of the abbreviations.

-       Review the use of abbreviations, for example, reactive oxygen species suddenly appears abbreviated and in other instances not.

-       The authors should consider abbreviating reactive oxygen species, which are mentioned throughout the article. For example, hydrogen peroxide, superoxide, hydroxyl radical, etc. Additionally, some are abbreviated while others are not.

-       Figures 5, 6, 7, and 8: Begin with an explanation of the figure, followed by the meanings of the abbreviations. Additionally, all abbreviations used should be cited in the figure caption.

-       Develop this idea: Mitochondria may also be involved in the generation of estrogen-associated ROS [267].

-        Figure 10 is cited twice in the same paragraph. I suggest citing it only once.

-       The conclusion seems more like a list of what the review contains rather than a conclusion. I suggest improving the wording and providing concise conclusions from the discussion.

Author Response

This manuscript is a resubmission of an earlier submission. The following is a list of the peer review reports and author responses from that submission.

Round 1

Reviewer 1 Report

Comments and Suggestions for Authors

The manuscript has a 49% match with other published papers/documents. It seems that most of the paragraphs were copied and pasted. 

The manuscript must be rejected.

Comments on the Quality of English Language

Moderate editing of English language required

Reviewer 2 Report

Comments and Suggestions for Authors

Authors present a work addressing: ‘'Oxidative stress in breast cancer’. Authors in this review systematizes information about the metabolic features of breast cancer directly related to oxidative stress. The topic of the article is interesting for clinical practice. However, the paper presents a few major issues including:

1. I believe that the title of the work is too short, it could be expanded to interest readers more.
2.  Authors should add following articles to the manuscript:
a) Napiórkowska-Mastalerz M, Wybranowski T, Bosek M, Kruszewski S, Rhone P, Ruszkowska-Ciastek B. A Preliminary Evaluation of Advanced Oxidation Protein Products (AOPPs) as a Potential Approach to Evaluating Prognosis in Early-Stage Breast Cancer Patients and Its Implication in Tumour Angiogenesis: A 7-Year Single-Centre Study. Cancers (Basel). 2024 Mar 6;16(5):1068. doi: 10.3390/cancers16051068.
b) Huang, Y.J.; Nan, G.X. Oxidative Stress-Induced Angiogenesis. J. Clin. Neurosci. 2019, 63, 13–16.
 c) Suvakov, S.; Jerotic, D.; Damjanovic, T.; Milic, N.; Pekmezovic, T.; Djukic, T.; Jelic-Ivanovic, Z.; Savic Radojevic, A.; Pljesa-Ercegovac, M.; Matic, M.; et al. Markers of Oxidative Stress and Endothelial Dysfunction Predict Haemodialysis Patients Survival. Am. J. Nephrol. 2019, 50, 115–125.
d) Chiang, F.F.; Chao, T.H.; Huang, S.C.; Cheng, C.H.; Tseng, Y.Y.; Huang, Y.C. Cysteine Regulates Oxidative Stress and Glutathione-Related Antioxidative Capacity before and after Colorectal Tumor Resection. Int. J. Mol. Sci. 2022, 23, 9581.
3. In section 5 some information regarding the CHEK2 and PALB2 genes and oxidative stress should be added.
4. I believe that a separate paragraph on angiogenesis and oxidative stress would enrich the article.
References:
a) Mdkhana B, Goel S, Saleh MA, Siddiqui R, Khan NA, Elmoselhi AB. Role of oxidative stress in angiogenesis and the therapeutic potential of antioxidants in breast cancer. Eur Rev Med Pharmacol Sci. 2022 Jul;26(13):4677-4692. doi: 10.26355/eurrev_202207_29192.
b) Shashni B, Nishikawa Y, Nagasaki Y. Management of tumor growth and angiogenesis in triple-negative breast cancer by using redox nanoparticles. Biomaterials. 2021 Feb;269:120645. doi: 10.1016/j.biomaterials.2020.120645.
c) Katary, M.A.; Abdelsayed, R.; Alhashim, A.; Abdelhasib, M.; Elmarakby, A.A. Salvianolic Acid B Slows the Progression of Breast Cancer Cell Growth via Enhancement of Apoptosis and Reduction of Oxidative Stress, Inflammation, and Angiogenesis. Int. J. Mol. Sci. 2019, 20, 5653. https://doi.org/10.3390/ijms20225653
5. Authors should strongly emphasize the unique character of the paper, why this review is important?, why their work is valuable in the oncology field?
6. In my opinion, Authors should add limitation of their literature analysis.

Minor
1. Modification of the grammar and punctuation is required.

Comments on the Quality of English Language

Modification of the grammar and punctuation is required.

Reviewer 3 Report

Comments and Suggestions for Authors

Comments on the Quality of English Language

The review is good, but it has so many errors and spelling mistakes. The authors should be asked to do a careful editing of the manuscript in the revision.

Round 2

Reviewer 1 Report

Comments and Suggestions for Authors

50% of overlap of the manuscript is unacceptable and a clear plagiarism. The manuscript must be rejected.

Comments on the Quality of English Language

Moderate editing of English language required

Reviewer 2 Report

Comments and Suggestions for Authors

Manuscript has been improved as suggested. In this way, it is suitable for publication.

Only two issues: 1. Please avoid abbreviation in the manuscript title (line 2).

2. Authors already explained the abbreviation for glutathione (GSH) in line 79, you don't need to repeat the same on lines 110, 250.

Reviewer 3 Report

Comments and Suggestions for Authors

There has been some mistake in providing the correct critique for the original version. I apologize for the error if it was mine. Please look at the critique for the current version to make revisions.

Comments on the Quality of English Language

Yes, the manuscript needs a little bit more careful editing for English.

Round 3

Reviewer 1 Report

Comments and Suggestions for Authors

A reject recommendation cannot be changed even after a revision of the manuscript. 

Comments on the Quality of English Language

Moderate english revision required.

Reviewer 3 Report

Comments and Suggestions for Authors

The authors have done their best to respond to the critique. There are still some minor issues that need to be addressed to make this manuscript final.

Line 94: The term “aerobic glycolysis” is used only to describe the Warburg effect in cancer cells where glucose gets converted to lactate even in the presence of oxygen. If the authors are talking about glucose metabolism in oxidative cancer cells where pyruvate does indeed go through mitochondrial oxidation, then remove the word “aerobic”.

Line 168: Change “In oncology” to “In cancer”. Oncology is a field of study.

Lines 170-171: NADPH is not used for oxidation of pyruvate into malate. I think that the authors are thinking about malic enzyme, which converts malate into pyruvate to generate NADPH.

Fig. 2: Correct the spelling for Lactate (three places).
